# Early onset of aridity in the past millennium: Insights from vegetation dynamics and climate change in the alpine, cold-desert region of Trans Himalaya, India

Ruchika B. Mohanty[1]*, Amit K. Mishra[2], Kriti Mishra[3], Akhilesh K. Yadava[2], M. Firoze Quamar[2,4], Iswar C. Barua[1], Ratan Kar[2,4]*

1 Department of Agronomy, Assam Agricultural University, Jorhat, India, 2 Birbal Sahni Institute of Palaeosciences, Lucknow, India, 3 Central Ground Water Board, North Western Region, Chandigarh, India, 4 Academy of Scientific and Innovative Research (AcSIR), Ghaziabad, India

* ruchikabajpayee@gmail.com (RBM); ratan_kar@bsip.res.in (RK)

**Data Availability Statement:** All relevant data are within the paper and its Supporting Information files.

## Abstract

Palynological analysis of surface soil and sub-surface sediments from the outwash plain of Hamtah Glacier, Lahaul-Spiti, India, has brought out the vegetation and climatic changes in the area during the last 1580 years. The arboreal and non-arboreal pollen ratio (AP/NAP) has been used to demarcate the different vegetation and climatic zones, complemented by the frequencies of the broad-leaved taxa. Lower values of thermophilous, broad-leaved arboreal taxa, indicate that the region experienced cold-arid conditions between 1580 and 1330 yr BP (AD 370–620); which can be related to the Dark Ages Cold Period (DACP). Thereafter, between 1330 and 950 yr BP (AD 620–1000), a rejuvenation of the broad-leaved elements reflects the initiation of a comparatively warm and moist phase, marking the Medieval Climatic Anomaly (MCA) in the region. The warm-moist phase was, however, short-lived, and from 950 yr BP to the Present (AD 1000 onwards), the region saw a return to cold-arid conditions, as evidenced by a sharp fall in the AP/NAP ratio. This cold-arid phase was, nevertheless, punctuated by a warm-moist period during 790 to 680 yr BP (AD 1160–1270), which marks the terminal phase of the MCA. After the termination of the MCA, the Little Ice Age (LIA) is well-marked in the area. The culmination of the long cold-arid regime is characterized by warmer conditions over the last 160 years, which is the manifestation of the Current Warm Period (CWP). Magnetic susceptibility ($\chi$lf) and sediment geochemistry (Weathering Index of Parker) were also attempted to have a multi-proxy approach, and show a general compatibility with the palynological data. The palaeoclimatic evidences suggest shorter warm periods and extended colder phases during the last 1580 years; in this high-altitude, cold-desert, Trans Himalayan region.

**Funding:** : RBM is grateful to the Department of Science & Technology (DST), New Delhi for financial assistance under the Women Scientist Scheme (SR/WOS-A/EA-15/2019) and RK thanks DST for sponsoring a project under the Climate Change Programme (DST/CCP/PR/07/2011/G) under which field work and sampling were done. The funders had no role in study design, data collection and analysis, decision to publish, or preparation of the manuscript.

**Competing interests:** The authors have declared that no competing interests exist.

## Introduction

The recent report of the Intergovernmental Panel on Climate Change (IPCC), shows an increase in mean global temperature by 1.5˚C from the pre-industrial times, and warns about the associated risk of droughts, heat waves and extreme rainfall events [1]. The evidences of warming are observed in the Himalayan region as well, with a rapid increase in temperature impacting the local ecosystems [2]. The precipitation records, in particular, reflect large scale spatial variations [3]. Therefore, an understanding of the spatial and temporal climate variability, under the influence of recent anthropogenic activities and past natural factors, is imperative to model future scenarios of climate change. As the instrumental data provides information only for the past century, or even lesser, for the remote Himalayan regions; the high-resolution, multi-proxy climatic records provide valuable information in this regard [4].

The Himalayan range has a great influence on the regional climate systems, and the snow and glaciers in the Himalaya play a key role in the societal and agrarian economy of the Indian subcontinent. The Himalaya is the most glaciated region on Earth outside the Polar realm, and is influenced both by the Indian Summer Monsoon (ISM) in the summers and by the Western Disturbances (WDs) in the winters [5]. The high-altitude, mountainous regions of the Himalaya are one of the most susceptible recorders of past climatic changes, as the climatic vicissitudes are well manifested in the glacial processes and deposits. The fluctuating intensities of the climate system (ISM and WDs) have had variedly impacted the Himalayan and South Asian regions. The precipitation pattern over the Himalaya is quite different; the fusion of the Higher or Greater Himalaya with the Pir Panjal Range in northwestern Himalaya forms a distinct climatic divide: the windward southern slopes receive heavy rainfall from the ISM, while the leeward northern part is the rain-shadow zone, where ISM rainfall is almost negligible. In contrast to the moist southern part, the northern Trans Himalayan region is a high-altitude, cold-desert, which receives heavy snowfall, through the WDs, during the winters [5].

It has been observed that the high-altitude alpine vegetation, on the threshold of climatic limits, is quite sensitive to climate change. Pollen and spores are one of the best tools to reconstruct the past vegetation, especially in the terrestrial realm, because of their abundance, good preservation and ubiquitous presence in the sediments. Even small amounts of sediments yield bountiful pollen-spores, which owing to their unique morphology, can be related to a particular species, genus or family, and can thereby be used to interpret the vegetation and contemporaneous climatic changes in a region, at a particular point of time [6–8].

Variations in the magnetic mineralogy of the sediments, which can be related to weathering, reflect changes in the climate. The transport and deposition of magnetic minerals during weathering are manifested by the changes in the magnetic susceptibility of the sediments; dependent nonetheless on the lithologies of the parent material. During warm-humid periods, the supply of magnetic minerals in to the sediments is enhanced due to weathering. Therefore, the phases of greater susceptibility, within the profile, can be used to identify the periods of climatic amelioration; whereas during cold-arid phases, weathering is reduced and thus, the susceptibility [9, 10].

Similarly, warm and moist conditions result in increased physical and chemical weathering; and the cold and arid phases are marked by the subdued nature of such processes. The stratified glacial deposits provide the required source material to interpret the temporal climatic oscillations [11]. The relative depletion and enrichment of major oxides can be interpreted in terms of weathering of the parent rocks [12]. The climate induced changes in the physical and geo-chemical processes are recorded by the sediments in the form of quantitative representation of the major and trace elements [13].

Anthropogenic less intervened high-altitude regions are excellent archives for studying the plant-climate interactions. In this context, the peri- and proglacial regions of the Himalaya, having endowed with a distinct alpine flora, are one of the most suitable sites to study the interface of vegetation and climate of the past. The primary objective of the present work is to provide information on the Late Holocene climatic changes, mainly on the basis of palynology, with additional inputs from mineral magnetism and geochemistry, from one of the high-altitude, cold-desert regions of the Indian subcontinent. Such studies from the glaciated parts of India are scarce, even more so from the arid Trans Himalayan region. The important climatic episodes of the last millennium are the Medieval Climate Anomaly (MCA), the Little Ice Age (LIA) and the recent Current Warm Period (CWP). However, it is important to note that the timing and duration of the above events, especially the MCA and the LIA, appear to be spatially asynchronous and require due attention.

## Regional setting

### Geology and geomorphology of the area

The Lahaul-Spiti District is situated in the northern part of Himachal Pradesh State and is one of the least populous districts of India. It consists of two formerly separate districts of Lahaul and Spiti (merged in 1960), with its administrative capital at Keylong (Fig 1). It is bounded by the Pir Panjal and Great Himalayan ranges towards its southern and northern extremities, respectively. The Great Himalayan Range disposes obliquely and merges with the Pir Panjal Range near the head of the Bara Shigri Glacier (5 km northeast of the present study area) and differentiates the water divide between the Higher and Trans Himalaya. It is drained by the Chandra and Bhaga rivers, which after their confluence at Tandi, form the Chandrabhaga (Chenab) River. The Lahaul Valley is approachable from the Rohtang Pass; it is about 16 km wide and extends from Kunzum La (pass to Spiti) to Baralacha La (pass to Ladakh) (Fig 1). The glacio-fluvial activities of the Quaternary Period have largely shaped the landscapes of the region [14].

Due to heavy snowfall from November to March, the Lahaul Valley remains practically cut off for over six months due to the closure of the passes, and that is why the area is one of the most inaccessible, though habited, regions of the Indian part of the Himalayas. The area exhibits a rugged topography with signatures of intense glacial activities, with well carved U-shaped glacial valleys and moraines being the most common features. The lithology and climate typically control the topography of the region. Hill slopes are mostly laden with frost shattered boulders and pebbles.

### Hamtah Glacier

The Hamtah glacier is about 6 km long, 0.5 km wide, the snout is at an altitude of 4020 m.a.s.l., from where the Hamtah *Nala* (stream) discharges and meets the Chandra River, about 7 km downstream. A series of recessional and terminal moraines just after the snout, reflect the rapid retreat of the glacier during the recent past. Calving of ice-blocks adjacent to the snout, dead ice-mounds and multiple serrac-faces, further point towards the degrading nature of the glacier (Fig 2A). The outwash plain extends to about 2 km and is marked by a characteristic U-shaped profile. Lateral moraines can be traced about 4 km downstream, however, kame-terraces are absent. Markings made by lateral moraines can be observed at higher levels along the valley walls, indicating the past extent of the glacier (Fig 2B and 2C).

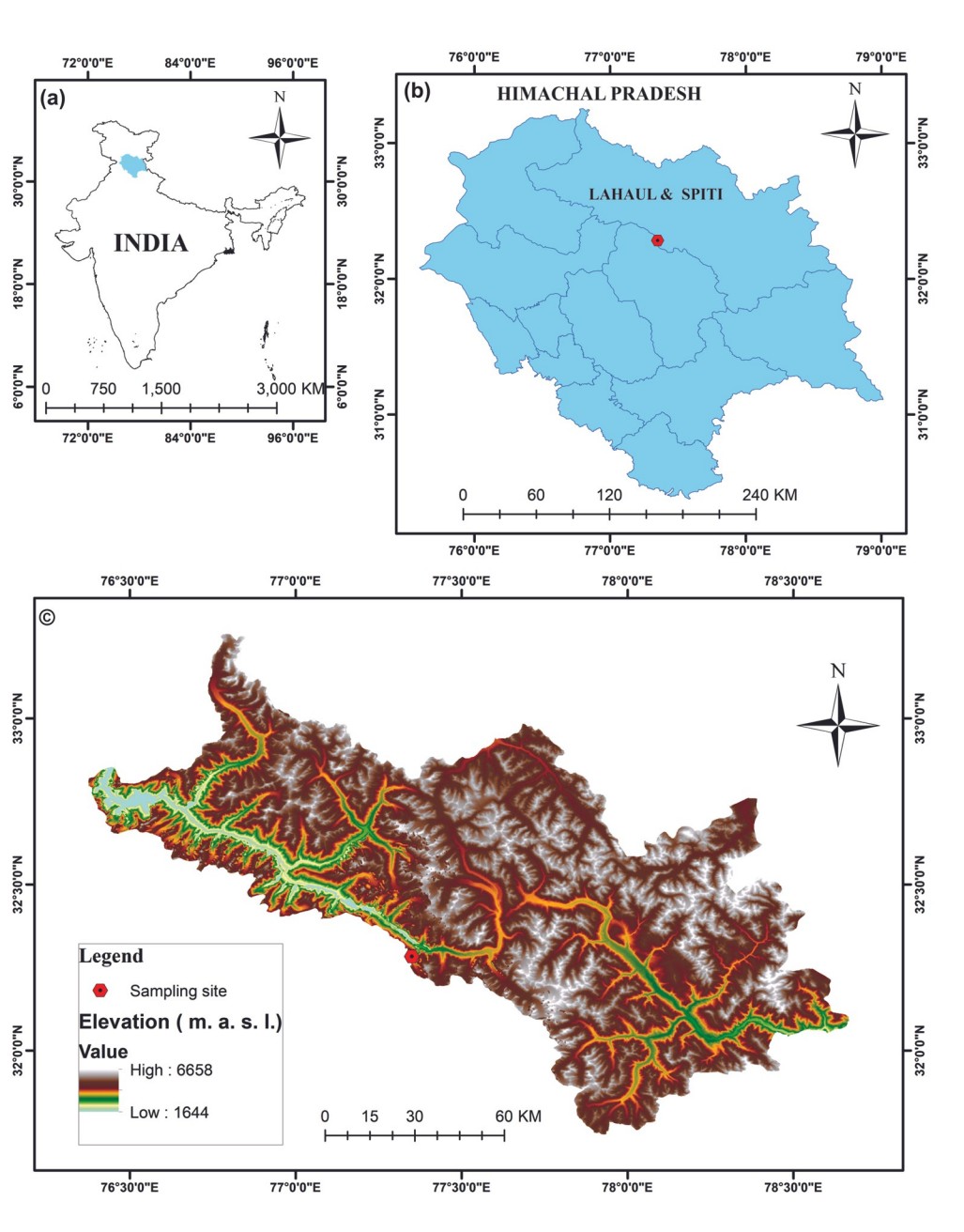

**Fig 1.** a Map of India showing the State of Himachal Pradesh, b. Map of Himachal Pradesh showing the sampling location in the Lahaul and Spiti District (red dot), c. Shuttle Radar Topographic Mission (SRTM) digital elevation map (DEM) of the Lahaul and Spiti District, Himachal Pradesh, India, showing the sampling location (red dot) around the study area (the figure has been made using ArcGIS 10.3).

## Climate and vegetation

The area of the present study, the Lahaul Valley, experiences a general cold-dry climate. The elevated Pir Panjal Range blocks the clouds of the ISM, hence the southern flanks of Pir Panjal Range receive high precipitation, while the area towards its north falls in the rain-shadow zone

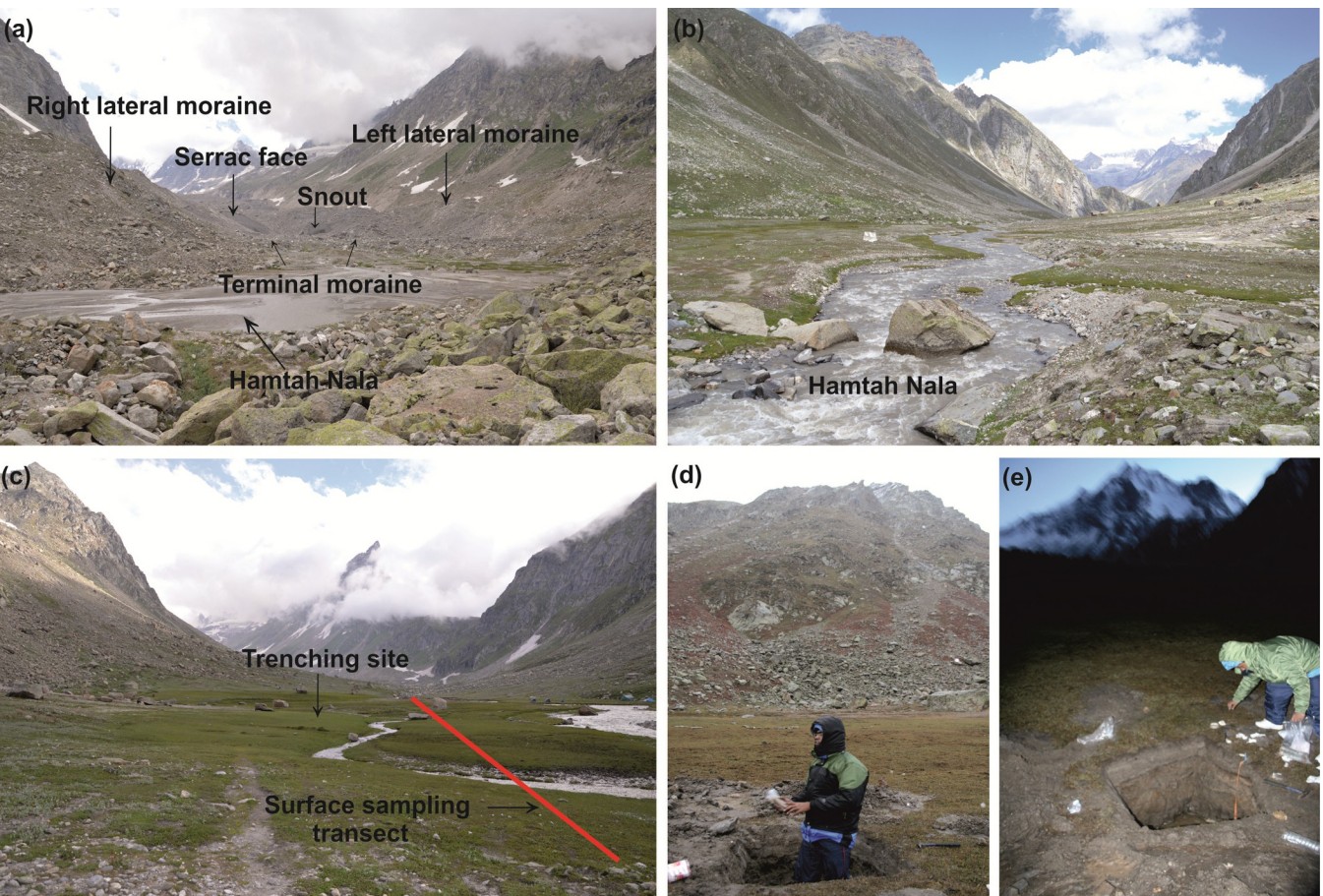

**Fig 2. Field photographs.** a. Different geomorphic features of the Hamtah Glacier, b. Downstream view of the outwash plain, c. Upstream view of the outwash plain, d–e. Trench for subsurface sampling in the outwash plain.

[14]. Sporadic, low intensity ISM showers occur from July to September, whereas most of the precipitation falls as snowfall from November to March due to the WDs. The average annual precipitation is only 150 mm, while the temperature is between −15°C to 28°C [15].

Based on the classification of the flora of India by Champion and Seth [16], the region falls under 'dry alpine forests'. In general, it is marked by sparse vegetation, characterized by patches of ephemeral herbaceous taxa. The vegetation growth starts in May-June, it reaches its climax during August and starts to deteriorate by September-October [17]. The outwash plain is an alpine meadow, characterised by a tuft of grasses (Poaceae). Scattered patches of members of Rosaceae, Asteraceae, *Artemisia*, Brassicaceae, Polygonaceae, Ranunculaceae, Euphorbiaceae, Ephedraceae, Papaveraceae and Convolvulaceae are present. During the summers, the outwash plain presents a lush green aspect, with flowers of different hues providing a colourful panorama. Besides the outwash plain, the surrounding mountainous slopes, lateral moraines, kame-terraces and cliffs are otherwise barren, and present a dry, rugged outlook. Arboreal elements are completely absent around the outwash plain. Nevertheless, isolated trees of *Betula*, *Ulmus* and *Alnus* can be observed, some 5 km downstream of the outwash plain, on the rugged mountain slopes. Conifers are completely absent around the glacier. However, lush coniferous forests comprising *Cedrus deodara*, *Abies pindrow*, *Pinus wallichiana* and *Picea smithiana* are present on the windward side of the Pir Panjal Range, up to an altitude of around 3500 m.a.s.l.

## Material and methods

### Field work

The present work is part of the in-house research projects of the Birbal Sahni Institute of Palaeosciences (BSIP), Lucknow, India; an autonomous Institute under the Department of Science and Technology, Government of India. The research area falls under Project number 6 of BSIP–'Late Pleistocene–Holocene vegetation and climate reconstructions for the Himalayan region: understanding the dynamics and forcing mechanisms.' The project is approved by the Research Advisory Council and the Governing Body of BSIP. The project includes conceptualization of the research problem, field work for sample collection, data generation and publication of the results. The Director, BSIP, approves the field work.

Prior to the collection of the sub-surface sediments, trial trenches were dug at a number of places in the area. It was observed that the outwash plain was the only geomorphic feature that afforded sediments of some thickness to enable palaeoclimatic studies. Consequently, a trench was dug on the outwash plain of the glacier (at ~4000 m.a.s.l.) up to a depth of 90 cm, after which further digging was not possible as morainic boulders were struck (Fig 2D and 2E). The wall of the trench was scrapped cautiously to avoid contamination of the sediments prior to sampling and eighteen samples were collected at an interval of 5 cm each. The samples were packed in air-tight plastic zipper sampling bags, assigned appropriate sample numbers and the details of the profile were noted in the field. The trench was named HOA ('HO' representing Hamtah Outwash and 'A' the number of the trench) and the samples have been named HOA-1 to HOA-18 from bottom to top.

### Laboratory work

**Chronology.**  The chronology for the sedimentary sequence has been obtained by $^{14}$C AMS dates of four samples. Of these, three samples at 5–10 cm, 25–30 cm and 80–85 cm were dated at the Poznan Radiocarbon Laboratory, Glewice, Poland and the one at 60–65cm was dated at Direct AMS, US. The dates were used as provided by the respective laboratories, which are 40±20 yr BP at 5–10 cm, 840±25 yr BP at 25–30 cm, 1395±33 yr BP at 60–65cm and 1605±20 yr BP at 80–85 cm (Table 1). The Bayesian age-depth model for the 90 cm trench was made by using the R package rbacon modeling program [18], which gives a chronology of ca. 1580 yr BP (Fig 3). Since the age of 40±20 yr BP (at 5–10 cm) is modern, hence it is not part of the age-depth model.

**Palynological analysis.**  The processing of the samples was done following the well established method of maceration [19, 20]. The pollen-spores were extracted from the sediments involving the step-wise processing of boiling 20 g of each sample in 10% KOH solution for 15 minutes to separate the pollen from the sediments, followed by treatment with 40% HF acid to remove the silica. Thereafter, to provide clarity to the pollen-spores, the important step of acetolysis was done by using an acetolysing mixture (1:9 concentrated sulphuric acid and acetic anhydride, respectively). The final macerates were preserved in vial tubes with 50% glycerin

**Table 1.** $^{14}$C AMS dates of the four trench samples.

| Sample No. | Depth (cm) | Laboratory | Laboratory No. | Sample type | Age $_{14}$C (yr BP) |
|---|---|---|---|---|---|
| HOA-17 | 5–10 | Poznan Radiocarbon Lab., Poland | GdA-3529 | Bulk sediment | 40 ± 20 |
| HOA-13 | 25–30 | Poznan Radiocarbon Lab., Poland | GdA-3528 | Bulk sediment | 840 ± 25 |
| HOA-6 | 60–65 | Direct AMS, US | D-AMS 032483 | Bulk sediment | 1395±33 |
| HOA-2 | 80–85 | Poznan Radiocarbon Lab., Poland | GdA-3526 | Bulk sediment | 1605 ± 20 |

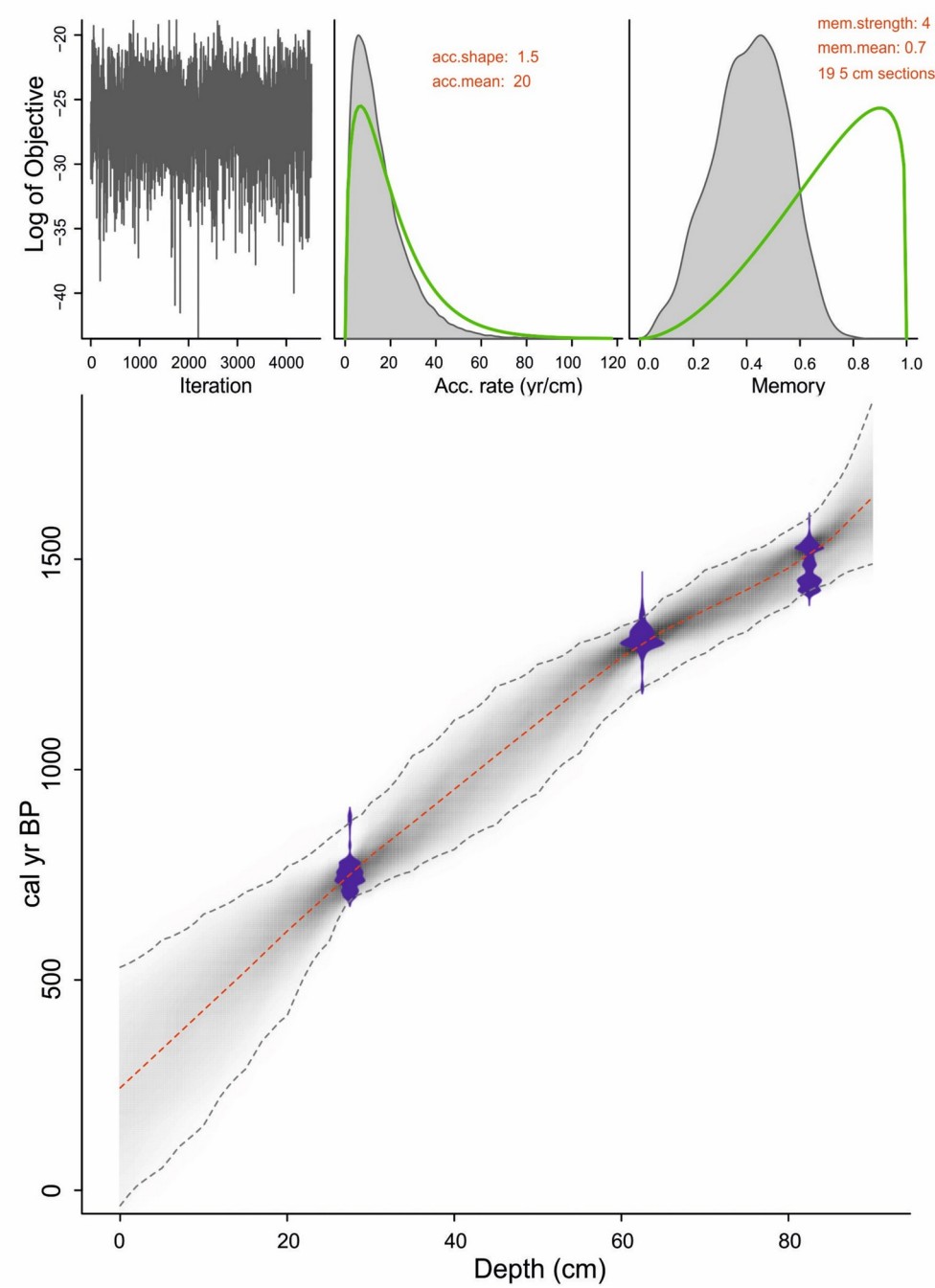

**Fig 3. Bayesian age-depth model of trench HOA.** The blue bars indicate the [14]C age distribution, the greyscale reflects the likelihood and the dotted red line follows the mean ages.

solution and two drops of phenol was added to inhibit post-maceration fungal growth. Microscopic study was done by making temporary slides mounted on glycerin, under an Olympus BX-61 light microscope with 40X magnification. Normally, around 400 pollen and spores were counted per sample, which denotes the Total Pollen Count (TPC). For deducing the arboreal pollen/non-arboreal pollen ratio (AP/NAP), only the respective percentages of each pollen taxa (phanerogams: flowering plants) are taken into account, which is taken as the Total

Pollen Sum (TPS), wherein the spores of cryptogams (fungi, algae, bryophytes and pterido-phytes) are excluded.

**Grain size analysis.** For grain size analysis, all the samples were treated with 2% HCl and 10% $H_2O_2$ to remove the carbonates and organic matter from the sediments. Grain size distribution of the eighteen sub-surface samples of the trench was determined with a multi-wave length Laser Particle Size Analyser (LPSA), Beckman Coulter LS$^{TM}$ 13 320. The Beckman Coulter LS$^{TM}$ 13 320 instrument offers the highest resolution, reproducibility and accuracy [21].

**Environmental magnetic measurements.** For measuring the magnetic susceptibility, samples were normally prepared by drying the sediments at room temperatures, and if oven dried, then not exceeding 35˚C, as changes in the magnetic mineralogy may occur at higher temperatures. Dry samples were then hand-crushed and disaggregated in a pestle and mortar, wrapped in cling foil and packed into 10 cc non-magnetic (styrene) cubes. Bartington MS2B susceptibility meter was used for measuring the magnetic susceptibility. MS2B sensor is provided with dual frequency systems, both low frequency (LF, χlf at 0.47 kHz) and high frequency (HF, χhf at 4.7 kHz) to assess the concentration of magnetic minerals in the samples [22]. Low frequency (LF) magnetic susceptibility of the sediments was measured to provide an assessment of the climatic variations in the region. All the measurements were taken in S.I. units and the susceptibility was expressed in $m^3$/kg.

**Geochemical analysis.** For geochemical analysis, the samples were first air dried and 10 g of each sample was powdered to a mesh size of 200 μm in Tema Mill. Pressed pellets were made from 5 g each of the powdered sample, using polyvinyl alcohol as a binding agent [23]. The major elements were determined by X-ray fluorescence, using Bruker S8 Tiger XRF Spectrophotometer. International reference standards were used to check the precision of the instrument and accuracy of the analytical methods. The accuracy of measurement for all major oxides (except $P_2O_5$ and MnO) is better than 5% and the precision is better than 2% [24]. Volatiles such as $H_2O$ and $CO_2$, and organic matter are included in the major element analysis and are expressed as Loss on Ignition (LoI). For calculating the LoI, about 5 g of each sample was ignited at 950˚C for 4–8 hours and then measured again to calculate the loss of volatiles and organic matter.

## Results and interpretations

### Palynology of the surface sediments: Pollen–vegetation relationship

The development of modern analogues, in the area of study, has been done by us from the surface soil samples that were collected in a linear transect from the outwash plain of the Hamtah Glacier [25]. This was primarily undertaken to interpret how well the surrounding vegetation is represented in the pollen records, or not. All the samples are characterized by an overall dominance of tree-taxa (arboreals) over the herbaceous taxa (non-arboreals), and exhibit a very high abundance of *Pinus* pollen (63–81%). *Abies* (3–10%) and *Picea* (1–8%) also show good representation amongst the other coniferous taxa. The common broad-leaved tree-taxa are *Ulmus* (1–3%) and *Alnus* (0.5–5%), whereas *Betula* and *Corylus* occur sporadically and *Rhododendron* is rare. Amongst the non-arboreals, the major contributing taxa include Lamiaceae (2–6%), Amaranthaceae (Chenopodiaceae: Cheno/Ams) (0.5–11%), Cichorioideae (Liguliflorae) (<0.5–6%) and Rosaceae (1–4%). Other than these, Euphorbiacecae (<0.5–2%), Ranunculacecae (<0.5–1.5%), Asteroideae (Tubuliflorae) (<0.5–1.5%) and Convolvulacecae (<0.5–1%) are present in moderate amounts. Members of Brassicaceae (0.5%), Apiaceae (<0.5–1%) and Polygonaceae (<0.5–1%) are recorded in low frequencies, while Ephedraceae is present only in one sample. Ferns spores are also less (0.5–2.5%), along with fungal and algal elements (Fig 4A and 4B and S1 File).

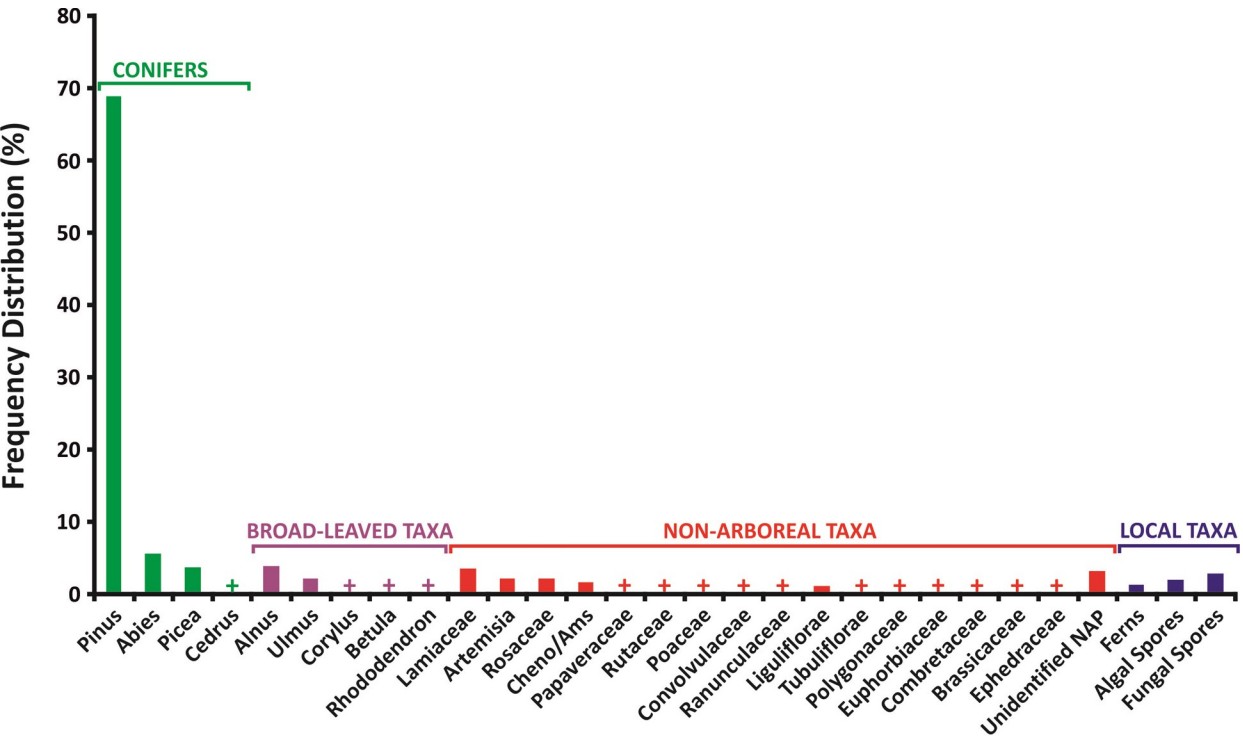

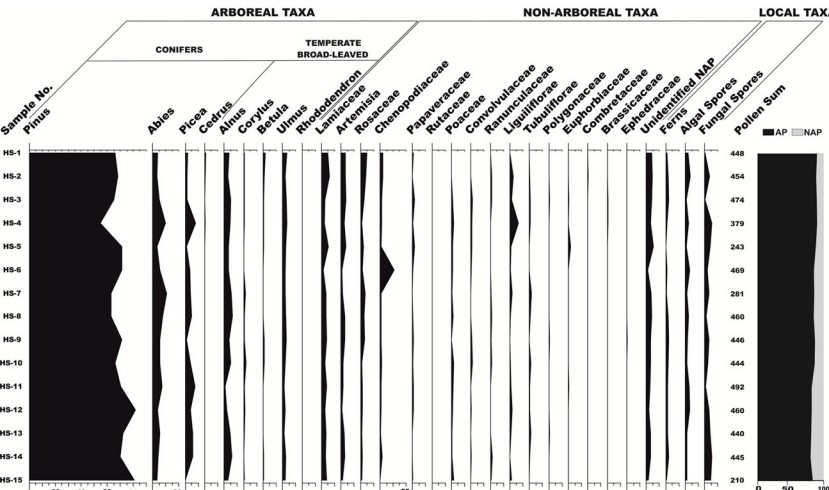

**Fig 4. a.** The collective frequency of different taxa from the surface sediments of the outwash plain, Hamtah Glacier (the frequency of different pollen taxa from all the samples have been put together for each of the taxa). **b.** Pollen spectra showing the frequency distribution of individual pollen taxa from the surface sediments of the outwash plain, Hamtah Glacier.

## Palynological analysis of the sub-surface sediments

Palynological analysis was undertaken on the samples of the 90 cm trench (HOA) that was dug on the outwash plain of the Hamtah Glacier. Throughout the sequence, a general preponderance of arboreal pollen is observed; nonetheless, subtle changes in the frequencies of different pollen taxa can be observed across the profile. From bottom to top, three pollen zones (Pollen

zone-I, Pollen zone-II and Pollen zone-III) have been demarcated in the sequence based on the increase or decrease in the percentages of arboreal pollen and non-arboreal pollen (here represented as ratio of arboreal/non-arboreal pollen). Besides, the changing pollen frequencies of the temperate broad-leaved tree-taxa, which are important climatic indicators, have also been taken into account in demarcating the respective pollen zones. The palynological composition of the three pollen zones, with respect to depth and chronology, are described below (Fig 5).

**Pollen zone-I (90–65 cm).** This zone spanning 250 years, between 1580 and 1330 yr BP encompasses a 25 cm sandy-silty layer. This zone represents the basal part of the profile, and shows a predominance of arboreals (81%), over the non-arboreals (19%). Among the arboreals, conifers represent 75%, while the broad-leaved taxa are 6% of the total assemblage. *Pinus* (41–71%) predominates over the others, along with *Picea* (2–13%), *Abies* (2–21%) and *Cedrus* (2–4%). The broad-leaved arboreals are mainly represented by *Ulmus* (1–3.5%), *Betula* (<0.5–1%) and *Corylus* (<0.5–1%), whereas, *Carpinus* is sporadically recorded in the assemblage. Among the non-arboreals, which represent the local herbaceous vegetation, Poaceae (1–4%), followed by Lamiaceae (0.5–3%) and Rosaceae (1–2%) are present in good frequencies, along with moderate values of Saxifragaceae (1–1.5%), Ranunculaceae (<0.5–2%) and Oleaceae (2%). Other taxa, such as Brassicaceae (<0.5–1%), Convolvulaceae (<0.5–1%), Papaveraceae (0.5–1.5%), Apocynaceae (1.5%), Combretaceae (<0.5–0.5%), Fabaceae (<0.5%) and Liliaceae (<0.5%) are in lesser values in the assemblage. Among the steppe elements, *Artemisia* (2–8%), Asteroideae (Tubuliflorae) (0.5–2%) and Amaranthaceae (0.7–1.4%) are present in good numbers. Other than these, Cichorioideae (Liguliflorae) (4%), Ephedraceae (<0.5–1.5%), Solanaceae (0.5%), Papilionaceae (0.5%), *Thalictrum* (<0.5%) and *Lonicera* (<0.5%) are present sporadically. Ferns (1–3%), along with fungal and algal remains, are consistently present in this zone (Fig 5 and S2 File).

**Pollen zone-II (65–40 cm).** This zone having a duration of 380 years, between 1330 and 950 yr BP, covers a 25 cm layer of fine-grained sand and silt. As in the previous zone, this zone is

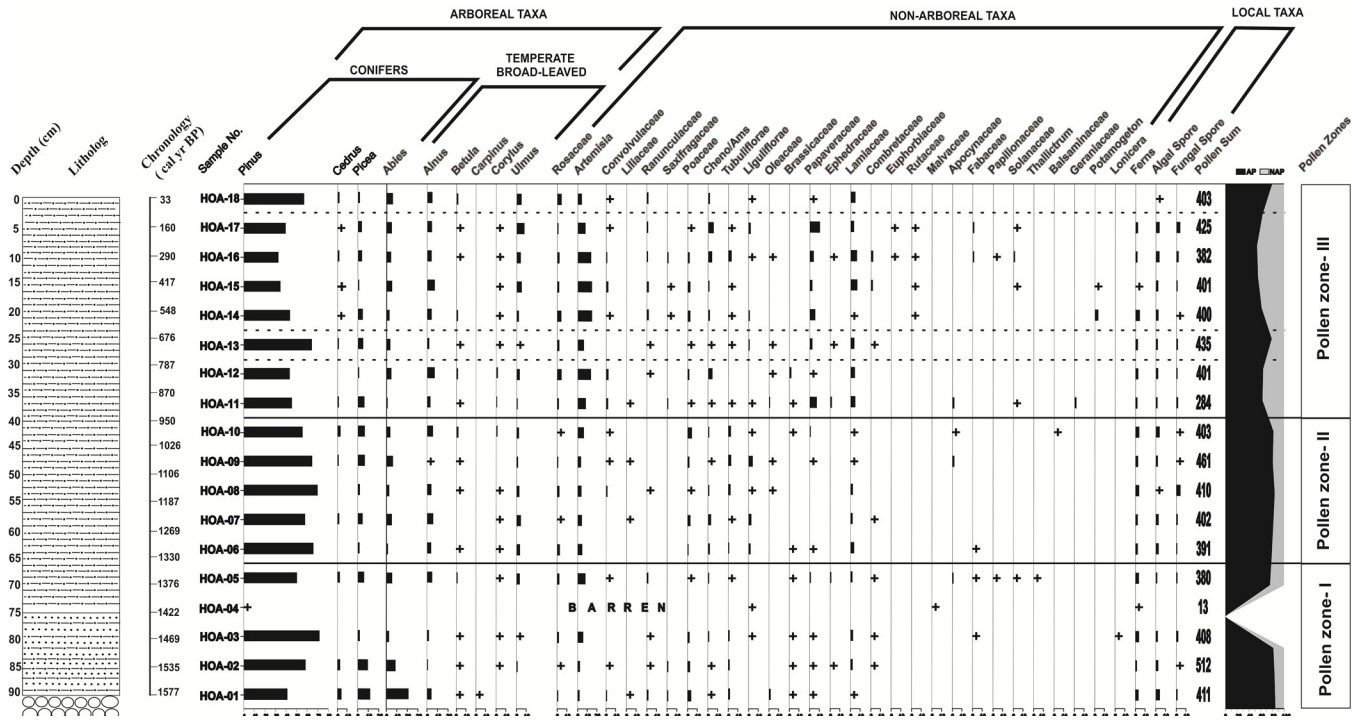

**Fig 5. Pollen spectra from the sub-surface sediments (trench) of the outwash plain, Hamtah Glacier.**

also marked by the preponderance of tree pollen taxa (82%) over the herbaceous elements (18%). An increasing trend in the AP/NAP ratio is observed at the initiation of this phase, followed by a decreasing pattern at its termination. *Pinus* maintains its predominance and ranges from 55–70%. *Abies* (1–6%), *Picea* (2–7%) and *Cedrus* (1–3%) are also well represented in good amounts. However, *Juniperus* is present only in one sample. This zone is, however, demarcated by an increase in the broad-leaved category (9%) as compared to the preceding phase. *Alnus* (0.5–6%) and *Ulmus* (1–4%) represent good frequencies, while that of *Betula* (0.5–2%) and *Corylus* (0.5–1%), though in low values, also show a subtle increase. Among the non-arboreals, the prominent taxa are *Artemisia* (3–6%), Poaceae (<0.5–4%), Amaranthaceae (0.5–3%), Asteroideae (Tubuliflorae) (1–3%), Chicorioideae (Liguliflorae) (0.5–4%), Lamiaceae (0.5–3%), Apocynaceae (0.5–2%) and Rosaceae (0.5–2%). Other taxa, such as Oleaceae (<0.5–0.5%) and Papaveraceae (0.5–1%) are present in lesser numbers. Presence of Liliaceae, Ranunculaceae and Combretaceae is very low and infrequent. Balsaminaceae is present only in one sample. Fern spores (2–3%) are present in fair amounts, along with fungal and algal elements (Fig 5 and S2 File).

**Pollen zone-III (40–0 cm).** This zone covers a time span between 950 yr BP to the Present and comprises a 40 cm layer of fine-grained sand and silt, with rootlets at the top of the sequence. This zone is initiated by a prominent decline in the AP/NAP ratio, as compared to the previous zones. Thereafter, it shows a constant pattern in the AP/NAP ratio. However, a prominent peak is observed in the middle, and it also culminates with a sharp increase in the ratio at the top. The average frequency of the arboreals is 65%, while that of the non-arboreals is 34.5%. The conifers show a sharp decrease in values (55%) as compared to the previous zone, while the broad-leaved taxa, more or less, maintain similar frequencies (10.5%). The marked decrease in the AP/NAP ratio is the characteristic feature of this zone. *Pinus* pollen, as in the previous zone, is represented by high values (32–64%), and other conifers, such as *Picea* (1–7%), *Abies* (1–6%) and *Cedrus* (1%) are also common. Among the broad-leaved arboreals, *Alnus* (2–7%) and *Ulmus* (1–8%) are quite common, while *Betula* (<0.5–1%) and *Corylus* (1%) are comparatively less. Among the non-arboreals, *Artemisia* (4–14%) is dominant and shows its highest numbers in this zone. Rosaceae (1–4.5%), Amaranthaceae (<0.5–5.5%), Papaveraceae (<0.5–10.5%), Lamiaceae (<0.5–7%), Apocynaceae (2%) and Geraniaceae (2%) are the other common herbaceous taxa. Other than these, Convolvulaceae (<0.5–2.5%), Poaceae (0.5–3%), Oleaceae (<0.5–1%), Ranunculaceae (<0.5–2%), Brassicaceae (1–2%), Ephedraceae (1%), Fabaceae (1%), Solanaceae (1%) and Saxifragaceae (1%) are also common. Occurrence of Liliaceae, Asteroideae (Tubuliflorae), Chicorioideae (Liguliflorae), Combretaceae, Rutacaeae, Euphorbiaceae and Papilionacaeae are low and sporadic. The occurrence of fern, algal and fungal spores are similar to the previous zones.

Though, this zone shows reduced AP/NAP ratio (65/34.5%), however, a notable feature is the sharp increase in arboreal pollen in one sample (30–25cm; 790 to 680 yr BP; AP/NAP ratio: 79.5/20%). Besides, the terminal part of the zone (5–0 cm; 160 to 33 yr BP) also shows an increasing trend in the AP/NAP ratio (80.5/19%) (Fig 5 and S2 File).

## Comparison between the surface and sub-surface palynological assemblages

A comparison of the surface and sub-surface palynological data, has revealed the basic similarities between them (Fig 6). All the surface samples show an overall majority of arboreal pollen taxa, especially *Pinus*, which is the most dominant (63%). *Pinus* spp. are prolific pollen producers and owing to their excellent buoyancy, are efficiently transported by winds to long distances. They also have good preservation potential in the sediments, which enables them to be bountifully recorded in the palynological assemblages. It has been observed that even if *Pinus*

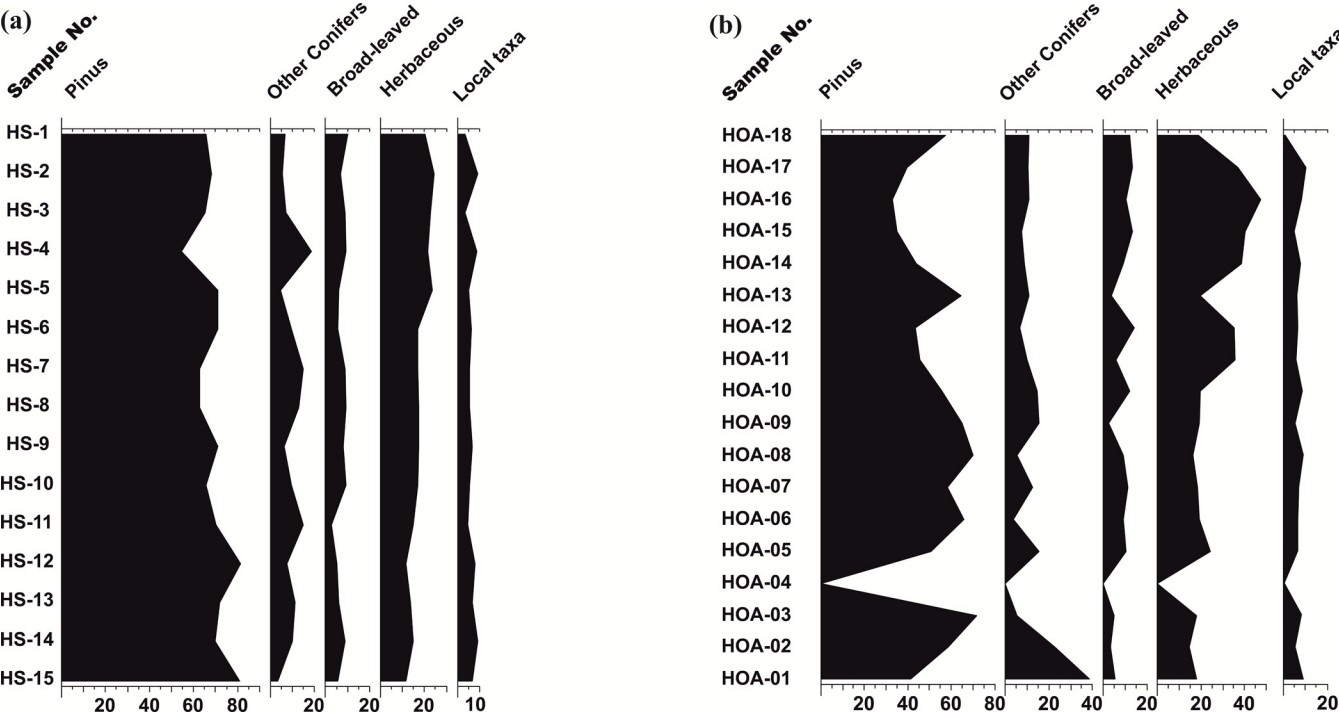

**Fig 6. Pollen assemblages of the major plant groups.** a. In the surface sediments, b. In the sub-surface sediments.

is not growing in the immediate vicinity of the sampling locations, its pollen can be predominant to such an extent that it completely overwhelms the other pollen taxa in the palynological records. A bountiful abundance of *Pinus* pollen has been observed in the surface sediments, from other glacial sites in the Western Himalaya, as well [26, 27]. Other conifers (*Abies*, *Picea* and *Cedrus*) constitute about 8.5% and the broad-leaved arboreals account for 6.5% of the pollen assemblage. The non-arboreal herbaceous vegetation shows an average value of 16% and the non-pollen palynomorphs (NPPs) account for the remaining 6% of the palynological assemblage. The sub-surface samples (HOA 1 to HOA 18) show a good similarity with the palynological records of the surface samples. The basic unity is the dominance of arboreal elements over the non-arboreals in all the surface, as well as sub-surface samples. Moreover, the high proportion of *Pinus* pollen, along with significant similarity in the palynological assemblages from both the realms has enabled the proper calibration of the sub-surface fossil pollen records for reconstructing the past vegetational changes around the study area. If the match between the two had not been undertaken, it could have created noise in the interpretation of the fossil pollen data, with regards to the absolute dominance of arboreal pollen, especially *Pinus*, across the sub-surface profile.

## Grain size analysis

The grain size parameters of the sediments (sand/silt/clay) can be used to interpret the depositional environment with respect to transportation and deposition [28]. In the present study, grain size analysis was primarily undertaken to corroborate the sedimentological data with the age-depth model (rate of sedimentation), and shows a general compatibility. Grain size data were obtained with the help of Gradistat Software [29]. According to the scheme of Ternary Diagram proposed by Folk and Ward [30], majority of the samples fall in the silt and silt-sand category, which also matches well with the field observations of the litholog (Fig 7).

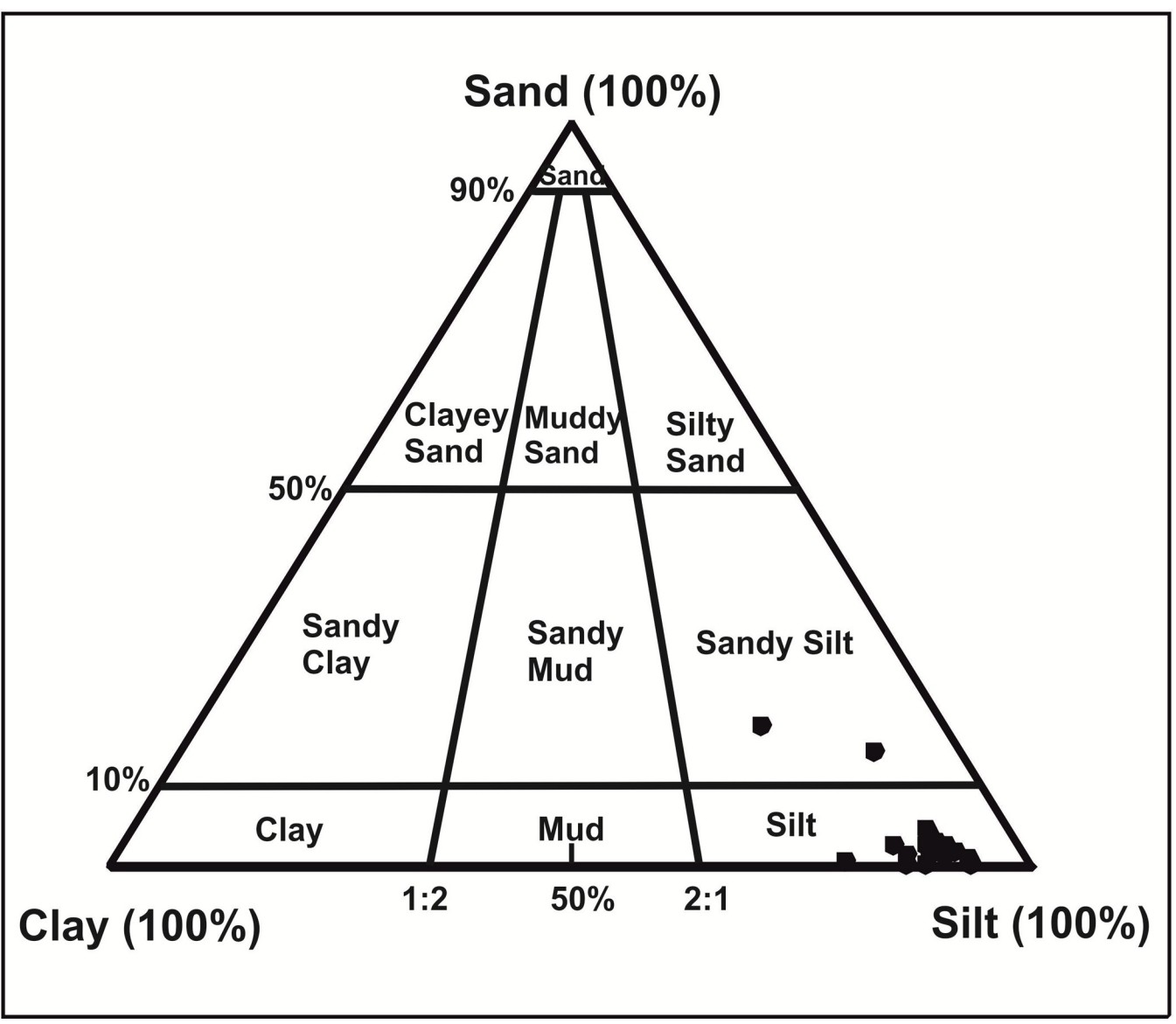

**Fig 7. Ternary diagram showing the percentage of sand, silt and clay in the sub-surface samples.**

## Magnetic susceptibility

Significant variations can be seen in the magnetic susceptibility ($\chi$lf) of the sediments across the profile, which range between 2.3 to 8.7 (S1 Table). The lowest value of $\chi$lf, is recorded at the depth of 35–40 cm, and the highest value at 10–15 cm. The lower part of the profile till the depth of 65 cm does not show much fluctuation in the values of $\chi$lf. Later on, in the upper part of the profile, there is a slight decrease in the $\chi$lf values from the preceding zone and subsequently, the values increase towards the top of the profile (Fig 8).

## Geochemical data: Major oxides

In the sub-surface sediments, $SiO_2$ varies from 57.62–79.28 wt % and is depleted towards the top of the profile. The $Al_2O_3$ ranges from 9.57–17.18 wt % and shows a negative correlation

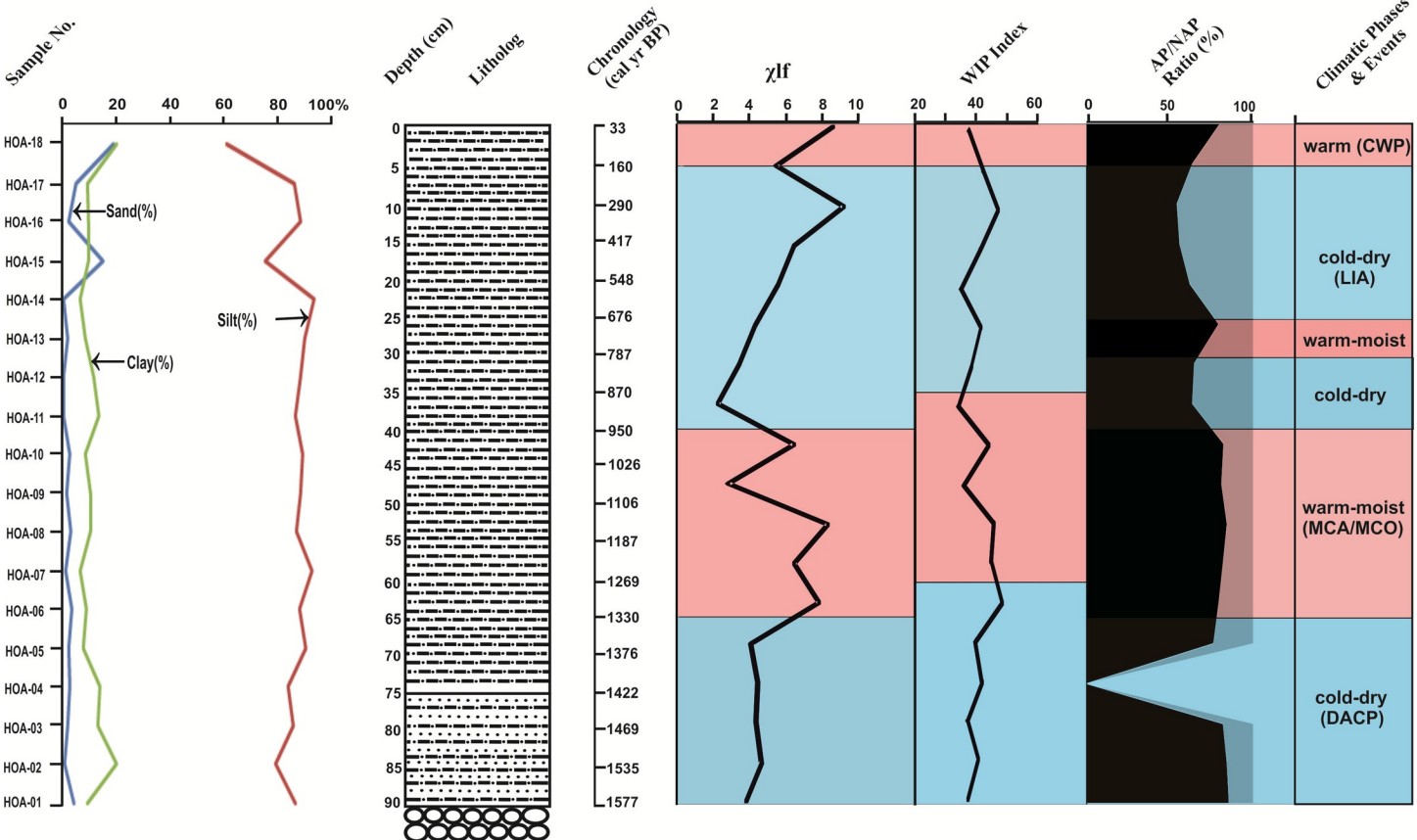

**Fig 8. Summary diagram showing the results of different proxies and the climatic phases.** (χlf: magnetic susceptibility, WIP: Weathering Index of Parker, AP/NAP: arboreal pollen / non-arboreal pollen ratio, DACP: Dark Ages Cold Period, MCA/MCO: Medieval Climatic Anomaly/Medieval Climatic Optimum, LIA: Little Ice Age, CWP: Current Warm Period).

with Si. $TiO_2$ ranges between 0.18 and 0.75 wt %. The values of $Fe_2O_3$ vary from 1.67–5.49 wt %, showing a negative correlation with Si. Highly mobile elements like Ca, Na and P show a positive correlation with Si. The values of CaO do not show much variation in its wt %. Similarly, $Na_2O$ and $P_2O_5$ do not show much variation in their values and show a positive correlation with Si (S3 File and S1 Fig).

The Weathering Index of Parker (WIP) has been used to ascertain the mobilization of elements during the process of weathering [12]. This is dependent upon the concentration of alkali and alkaline earth elements (Na, K, Ca and Mg), which are the most mobile of the major elements. The observed variations in the values of weathering indices throughout the trench reflect the changes in weathering intensity. The higher the values, lesser the weathering of the rocks. The WIP index falls between 47.8 and 33.5, which reveal the extent of chemical alteration in a proglacial environment (Fig 8).

## Discussion

### Palaeovegetation, palaeoclimate and correlation with respect to the Indian Himalaya

**1580 to 1330 yr BP.** The time interval of 1580 to 1330 yr BP, is marked by a AP/NAP ratio of 81/19%. The vegetation composition represents higher values of conifers: *Pinus*,

*Cedrus*, *Abies* and *Picea*, amongst which *Pinus* is having the highest frequencies (41–71%). The broad-leaved arboreals comprising *Corylus*, *Betula*, *Ulmus* and *Alnus*, however, have fewer values. Moisture loving elements, like pteridophytic spores, are scarce. This vegetation, which is best reflected in the decreased values of broad-leaved elements, suggests that cold-dry climatic conditions existed in the region during that time. The presence of *Artemisia*, a prominent high-altitude steppe element, in good amounts, also points towards low moisture availability. With the onset of the colder conditions, the thermophilous broad-leaved forests had declined; while on the other hand, the cold-tolerant coniferous forests, and the steppe-ground vegetation continued to flourish. The magnetic susceptibility values are somewhat compatible with the palynological data, though subtle variations between the two can be observed. The low values of $\chi$lf in the bottom part of the profile around 1580 to 1330 yr BP, as compared to the values in later part, suggest that during this period the area was under the influence of a cold and dry climate. The geochemical data are also, more or less, synchronous with the mineral magnetic and palynological data. The lower part of the trench, between 1580 and 1270 yr BP, records higher WIP values, which indicate that the sediments were subjected to lower rates of weathering during that phase. This cold phase recorded between 1580 to 1330 yr BP is the initiation of a colder period: the Dark Ages Cold Period [31]; after probably the culmination of the Roman Warm Period (2500–1600 yr BP) [32], which could not be recorded in the profile due to unavailability of archives older than 1580 yr BP (Fig 8).

The above short cold-arid phase is, so far, not well-marked in other Himalayan regions, though it is broadly comparable to some records in the Trans Himalayan region. Palynological data from the Tso Kar Lake, Ladakh, suggest that an expansion of Chenopodiaceae-dominated steppe vegetation after 4.8 kyr BP, indicate a sudden shift towards aridity in the region. A further weakening in the ISM is possibly indicated by the scarce vegetation cover between 2.8 and 1.3 kyr BP [33]. Around the Takche Lake, Lahaul-Spiti, a dominance of grasses and sedges is observed under a cold-dry climate ca. 2 ka, and this phase continued till around 1 ka [34]. From the Kunzum La (Pass), Lahaul-Spiti, palynological studies indicate the prevalence of alpine-steppe vegetation around 2–1.8 ka BP, under a cold-dry climate [35]. Another study from Kunzum Pass, at Sitikher bog, indicates that the glaciers had advanced in the region under cold climatic conditions during 2300–1500 yr BP [36]. From the Eastern Himalaya, lesser numbers of palynological data are available; nonetheless, a study from Jore-Pokhari area in Darjeeling states that between 1600 and 1000 yr BP, the climate turned cooler, resulting in the reduction of broad-leaved tree-taxa and an increase in the conifers [37]. Similarly from the Kupup Lake, Sikkim, drier climatic conditions have been inferred between 1800 and 1450 yr BP, which is reflected in an increase in the herbaceous taxa, and a consequent decrease in broad-leaved arboreals [38].

**1330 to 950 yr BP.**   During the subsequent phase between 1330 and 950 yr BP, the overall AP/NAP ratio remains almost the same (82/18%), as the preceding zone. This zone is, however, distinguished by an increase of the broad-leaved taxa, especially *Alnus* and *Ulmus*, which record a perceptible rise in the pollen spectra. The pollen assemblage further suggests the prevalence of relatively more diverse vegetation, with improved frequencies of fern spores, algal and fungal remains. The changing vegetation pattern, especially in the latter part of this phase, suggests an amelioration in the climatic conditions towards a warmer phase, as manifested by the rising trend of thermophilous broad-leaved elements, and the subtle reduction of conifers. This is also corroborated with corresponding high magnetic susceptibility values, depicting a transition towards warmer and moist conditions. A high in the $\chi$lf values indicates increased fresh water influx during the warm-moist phases. The WIP values show a decline between 1270 and 870 yr BP, giving a picture of enhanced weathering, influenced by relatively moist conditions during that time period (Fig 8). Moreover, the culmination of the MCA at around

790 to 680 yr BP is well recorded in the palynological assemblages. Besides, a period of aridity within the MCA, between 950 and 790 yr BP, is discernible in the present study.

This warm-moist phase has been recorded from other parts of the Western Himalaya, though, with some differences regarding its duration. This phase can be broadly correlated with the Medieval Climatic Anomaly (also called the Medieval Climate Optimum: MCO), which is a globally recognized warm event occurring around ca. AD 800 to 1200 [39, 40]. Around Sitikher bog, the glaciers retreated, with a corresponding shift of the tree-line towards higher elevations, under warm and moist conditions between 1500 and 900 yr BP [36]. From Naychhudwari bog, Lahaul-Spiti, an ascendency of the tree-line to higher elevations is also observed between 1300 and 750 yr BP [41]. The maximum expansion of thermophilous broad-leaved forests from the Chandra Valley, around Kunzum La, during 1609–1303 yr BP has been correlated with the MCA [42]. An expansion of the broad-leaved taxa around Triloknath glacier valley, Lahaul, suggests that the region witnessed a relatively warm climate between 2228 and 962 yr BP. [43]. From the ISM influenced Greater Himalaya, in Bhujbas near Gangotri Glacier, the abundance of local elements (*Betula* and *Juniperus*) and extra-local pollen (*Pinus*) between 1700 and 850 yr BP, indicates an amelioration of the climate, partly correlating with the MCA [44]. Around the Tipra Bank Glacier, broad-leaved tree-taxa *(Quercus, Betula, Alnus* and *Rhododendron)*, along with conifers *(Pinus, Abies* and *Picea)* occurred prior to 720 yr BP under warm-moist conditions [45]. A relatively warm and moist climate was suggested in the Pindari Valley, Kumaon Himalaya, between 1.75 and 0.9 kyr BP on the basis of the abundance of both broad-leaved taxa and conifers [46]. In the sub-tropical Lesser Himalayan region, the warm phase is better manifested with the maximum development of oak forests between 1400 and 400 yr BP, from the vicinity of Dewar Tal [47]. From the Eastern Himalaya (Sikkim), around Kupup Lake, a cold and moist phase has been suggested during 1450–450 yr BP [38]. From the Paradise Lake, Arunachal Pradesh, around 1100 yr BP, a diversity of conifers and broad-leaved vegetation is observed, which could be indicating a comparatively warmer climate [48].

**950 yr BP to the present.** From around 950 yr BP to the Present, a sharp fall in the AP/NAP ratio is recorded (65/34.5%), which distinguishes this phase. This is further accompanied by a prominent rise in *Artemisia* pollen, which records its highest frequencies during this period. Fern spores have also marked their presence in fewer amounts, than the preceding zone. The overall palynological assemblage suggests the prevalence of a cold and arid environment. The conditions were favorable for the growth of steppe elements and besides *Artemisia*, Rosaceae and Amaranthaceae are recorded in good frequencies. Other evidences like low magnetic susceptibility ($\chi$lf) indicate little transport of detrital material, as a consequence of less weathering. The values of geochemical indices consecutively increase, which implies that the area was under cold-arid conditions, which did not allow much chemical weathering (Fig 8).

After the culmination of the MCA, cold-arid conditions prevail in the region. The cold-arid conditions during this period, mark the onset of the LIA, which match well with the globally recorded cold phase during ca. AD 1500 to 1850 [49]. Moreover, an extended colder regime is witnessed in the region. However, within this overall cold-arid phase over the last 950 years, the vegetation reconstruction has revealed two distinct short-term warming trends. Between 790 and 680 yr BP (AD 1160 to 1270), there is a perceptible rise in the AP/NAP ratio (79.5/20%), which is indicative of a warm-moist hiatus. This warm phase is the manifestation of the culmination of the MCA. Thereafter, the terminal part of the zone (5–0 cm; 160 to 33 yr BP) is marked by an increasing trend in the AP/NAP ratio (80.5/19%), which is a good reflection of the present day warming (Current Warm Period: CWP; AD 1800 to the Present) [50]. The magnetic susceptibility and geochemical data also record this recent warming and are, more or less, synchronous with the palynological data (Fig 8).

This cold-arid phase of the LIA has been well recorded in the palynological studies from other places in the Himalaya, especially in the Trans Himalayan region. In Kunzum La, an expansion of dry steppe elements is observed from ca. 500 years to the present, indicating a drier climate [35]. A similar proliferation of steppe taxa is observed from ca. 400 years onwards around Takche Lake [34]. Studies from the other alpine regions of Lahaul-Spiti reveal that at Sitikher bog, the glaciers advanced and the tree-line descended to lower altitudes since 900 yr BP, under a colder regime [36], and around Naychhudwari bog the glacier advancements are observed over the last 450 years [41]. From the Chandra Tal region, a flourishing of meadow vegetation is discernible between 647–115 yr BP [42]; and a decrease in the broad-leaved tree-taxa around Triloknath Glacier since the last 300 years [43] correspond to the LIA. From the Greater Himalaya as well, the LIA event has been well marked, especially in the glacial valleys. In the Gangotri Glacier, a change in the vegetation is observed around 850 yr BP, with a distinct increase in the steppe taxa (*Ephedra* and other members of Asteraceae), accompanied by a decrease in the moist-loving elements (ferns and *Potamogeton*), indicating a trend towards drier climatic conditions [44]. From the Lesser Himalaya also, cold-dry conditions are discernible in the palynological studies. Around the Mansar Lake of Jammu province, pine forests came into prominence and replaced the oak dominated forests around 750 yr BP to the Present, indicating an onset of relative cold-dry phase in the region [51]. From Dewar Tal, Garhwal, a decline in oaks has taken place over the last 400 years, under a cold and dry regime [47]. From Eastern Himalaya, a study from the Paradise Lake [48], Arunachal Pradesh, indicates that the climate turned less moist around 550 yr BP, as manifested by a decline in the values of *Quercus*, *Tsuga* and *Juniperus*, corresponding to the LIA. From the Ganga Lake, Itanagar, the LIA is well marked, though, punctuated by intermittent warmer phases [52]. From Kupup Lake, Sikkim, deterioration in the climate has been reported during 400–200 yr BP [38]. The comparison of palynological data suggests that within a general similarity among the above climatic phases from across the Himalaya, the vegetation pattern and contemporaneous climate is asynchronous for individual sites, possibly due to altitude, topography and regional factors.

## Comparison of the present study with tree-ring records from the adjoining areas

As tree-rings provide one of the best resolutions of past climatic changes, the present work has been correlated with the available tree-ring data from the adjoining areas. The boreal spring precipitation records from the adjoining region in the Western Himalaya extending back to AD 1030 [53] indicate long term variability in extreme wet and dry events. The influence of LIA, in the Westerlies (WDs) dominated Trans Himalaya, recorded in the present palynological study, is broadly compatible with the tree-ring inferred temperature and precipitation records from Lahaul-Spiti and Kinnaur regions of Himachal Pradesh, respectively. Tree-ring inferred temperature records from the cold-arid, Lahaul-Spiti region have revealed a warm period from AD 940–1500 [54]. However, within this phase, the palynological records of the present study, show a shorter period of warming from 790–680 cal yr BP (AD 1160–1270), which, nonetheless, is also the warmest period as revealed in the tree-ring records. The cooling trend in the tree-ring based temperature records are discernible since the 16[th] century, which continued up to the 19[th] century, reflecting the cold phase of the LIA in the Lahaul-Spiti region. This is also consistent with the pluvial phase, as revealed by enhanced spring precipitation, in the adjoining Kinnaur region. The wet conditions reflected in the spring precipitation records from the cold-arid region are largely due to heavy snowfall under the influence of Western Disturbances [53], which is also manifested in the present palynological study as signatures of the influence of the LIA (Fig 9).

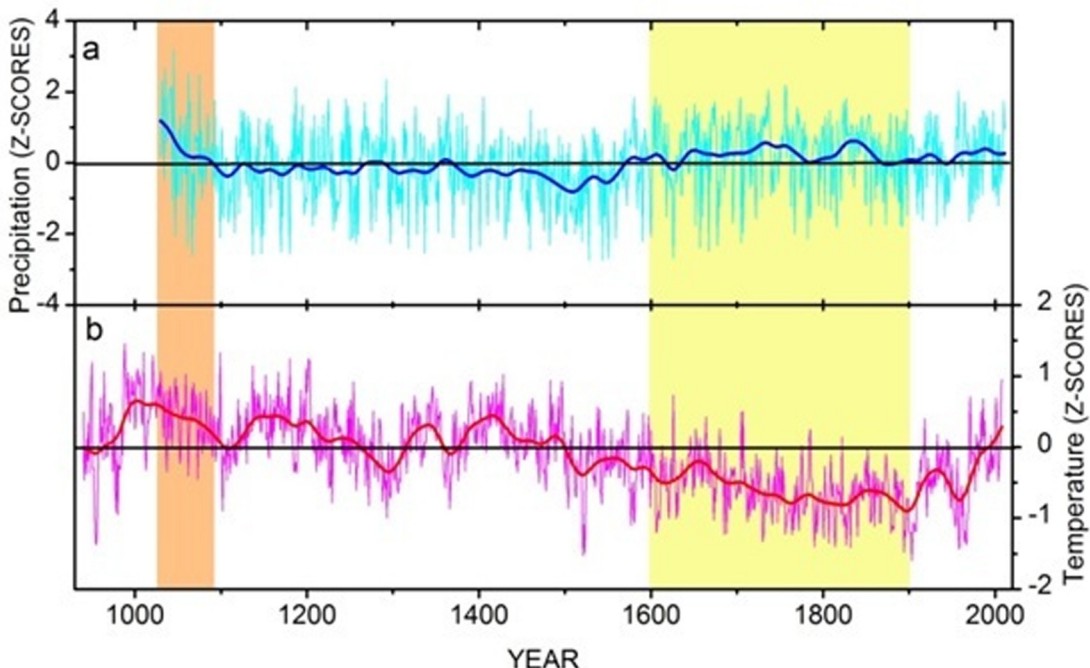

**Fig 9. Tree-ring based precipitation and temperature records from adjoining areas.** a. Boreal spring precipitation reconstruction (MAM) from Kinnaur, extended back to AD 1030 (Yadava et al. 2016), b. Temperature reconstruction (MJJA) from Lahaul and Spiti, dating back to AD 940 (Yadav et al. 2011). The red and blue lines are low pass filter showing the fluctuations in timescale of 50 years and above. The vertical orange bar indicates the moist condition in late 11[th] century, which reflects the MCA/MCO and the vertical yellow bar represents the influence of the LIA.

## Regional and global contextualization with respect to the Late Holocene climate events

The present work provides insights into the vegetation response to the climatic fluctuations during the Late Holocene from an alpine, cold-desert Trans Himalayan region, whereby four climatic events: the DACP, MCA, LIA and CWP were deciphered. However, a marked asynchronicity has been observed regarding the inception and duration of the above events from India and also from the different parts of the globe (Fig 10). The present study demonstrated colder and arid conditions between 1580 and 1330 cal yr BP (AD 370–620), which can be partially correlated with the Dark Ages Cold Period, globally known between 1600 and 950 cal yr BP (AD 350–1000) [31]. A multi-proxy study based on organic geochemistry (TOC and TN), stable isotopes ($\delta^{13}C$ and $\delta^{15}N$) and grain size from Himachal Pradesh, suggested reduced ISM rainfall between AD 450 and 950 [55]. A weakening of the ISM was inferred between 1550–1250 years BP, from the Bay of Bengal (India), on the basis of higher $\delta^{18}Ow$ values and was correlated with the DACP [56]. The expansion of sea-ice between AD 400–765, known as the North Atlantic ice-rafting event: Bond Event [57], is considered as the probable cause of the DACP. A weaker solar influence is also observed during this period [58]. Helama et al. [31] reviewed 114 palaeoclimatic studies and inferred that the DACP broadly ranges from AD 400–765 (1185–1550 cal yr BP).

Subsequently, between 1330 and 950 cal yr BP (AD 620–1000), warm-moist climate has been deduced around the study area. This phase of climatic warming is comparable with the MCA, which has been observed globally between AD 740 and 1150 [39, 40]. However, Lamb [59] and Crowley and Lowery [60] suggested the time-period of the MCA as AD 900–1300.

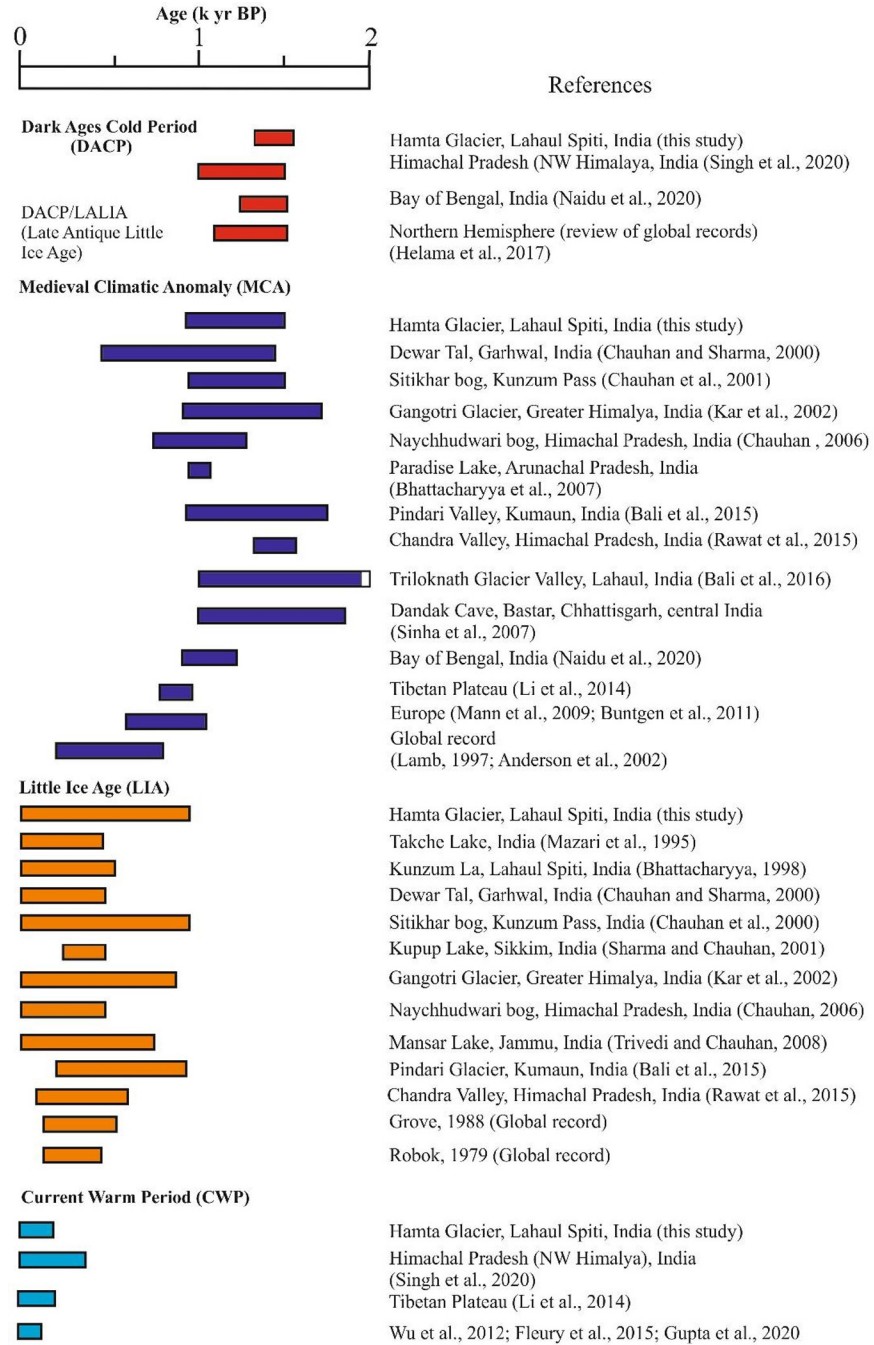

**Fig 10. The timing and duration of the Late Holocene climatic events from the Himalaya and other regions.**

The MCA was caused by the northward movement of the Intertropical Convergence Zone (ITCZ) [61]. In Europe, the warmer conditions during the MCA occurred between ca. AD 900 and 1300 [62, 63]. Gupta et al. [64], based on *Globigerina bulloides* records off Oman, suggested stronger Southwest Monsoon winds during the MCA. Based on δ$^{18}$O of speleothems from central India, the MCA was suggested to be during ca. AD 920–1300 [65]. Naidu et al. [56] observed negative δ$^{18}$Ow anomalies, and opined that the Bay of Bengal experienced

highest ISM rainfall during 1200–800 yr BP. Li et al. [66] suggested higher values of TN, TOC, magnetic susceptibility, sand and coarse silt in the sediments of Basomtso Lake, southeastern Tibet, which indicated more sediment input due to an increase in melt-water, due to warmer conditions during AD 1080–1140,.

Finally between 950 cal yr BP to the present (AD 1000 onwards), a prolonged cold and arid phase is discernible around the study area, which encompasses the LIA (AD 1440–1850) [49]. Robock [67] and Bradley and Jones [68] have suggested a broad time frame of the LIA as AD 1550–1850. The LIA could be linked to the strengthening, as well as the migration, of the Asian Jet Stream to the lower latitudes, causing the intensification of the WDs and resulting in colder conditions of the LIA over the Indian subcontinent [69]. There is a general consensus that the southward shift of the ITCZ has resulted in the weakest phase of the ISM across the Indian subcontinent during the LIA, and also during the past three-four millennia [70–75].

Meanwhile, within this extended cold and arid phase, a short pulse of warm-moist climatic fluctuation is noticeable between 790 and 680 cal yr BP (AD 1160–1270), which probably reflects the culmination of the MCA. Another warm climatic phase is discernible over the last 160 years, which can be well correlated with the Current Warm Period (CWP) [50, 76, 77]. Based on geochemical studies ($\delta^{13}$C and $\delta^{15}$N, TOC and TN), the CWP has also been recorded from Himachal Pradesh (NW Himalaya), during ca. AD 1600–2000 [55]. Higher values of TOC, TN, magnetic susceptibility, sand and coarse silt in the sediments of Basomtso Lake, southeastern Tibet, due to higher melt-water run-off owing to warmer conditions during AD 1790–2012, corresponds well with the CWP [66]. The increase in rainfall during the CWP has been linked to an increase in industrial emissions and other anthropogenic factors, as well as the associated increase in surface evaporation and convection in the Indian Ocean [40, 70, 78]. Zhou et al. [79] suggested that both the MCA warming, and the LIA cooling appear to be global events; however, the MCA was not as warm as the CWP, and that the LIA was more intense over the Eurasian continent. On the other hand, coral records from the South China Sea indicate that the climate of the MCA was similar to that of the CWP [80].

## Conclusions

1. Palynological analysis of the surface sediments (pollen–vegetation relationship) has revealed a predominance of *Pinus* pollen, which is due to its bountiful production and efficient wind dispersal.

2. The surface pollen data were found to be altogether compatible to the sub-surface fossil pollen assemblages. The development of the modern-analogues has helped in the calibration of the fossil pollen assemblages and the proper deduction of past vegetational and climatic changes in the region.

3. Lesser frequencies of broad-leaved taxa, low magnetic susceptibility and increasing values of WIP are observed around 1580 to 1330 yr BP (AD 370–620), indicating severity of climate that caused the degradation of broad-leaved taxa, and can be correlated with the DACP.

4. The initiation of a warm and moist period is observed between 1330 and 950 yr BP (AD 620–1000), which represents the MCA. The rejuvenation of broad-leaved elements, diverse herbaceous taxa, and improved values of the moist-loving ferns as well, are also compatible with increased magnetic susceptibility and lower values of WIP.

5. Around 950 yr BP till the Present (AD 1000 onwards), the region saw a return to general aridity; as evidenced by a sharp fall in the AP/NAP ratio, accompanied by a rise of the dry,

steppe elements. This cold-arid phase was, however, punctuated by a distinct warm-moist period between 790 and 680 yr BP (AD 1160–1270), which represents the terminal part of the MCA in the region.

6. After the culmination of the MCA, the complete extent of the LIA is well evidenced in the study area. Moreover, the palaeoclimatic data suggests the prevalence of extended colder phases, with an early inception of aridity in the last millennium, in this high-altitude, cold-desert, Trans Himalayan region.

7. Over the last 160 years, the vegetation is marked by an increasing trend in the AP/NAP ratio, which is the expression of the CWP. The magnetic susceptibility and geochemical data are broadly compatible with the palynological records.

8. The timing and duration of the Late Holocene climatic events, manifested in the different regions, are generally asynchronous.

## Supporting information

**S1 File. Spreadsheet showing the respective percentages of spore-pollen taxa in the surface samples.**
(XLSX)

**S2 File. Spreadsheet showing the respective percentages of spore-pollen taxa in the trench samples.**
(XLSX)

**S3 File. Geochemical data showing the major oxides (in wt %) of the trench samples, along with calculated values of the Weathering Index of Parker.**
(XLSX)

**S1 Table. Table showing the details of magnetic susceptibility in the trench samples.**
(PDF)

**S1 Fig. Binary plots of major oxides (in wt %) of the trench samples.**
(JPG)

## Acknowledgments

We thank the Director, Birbal Sahni Institute of Palaeosciences, Lucknow, India, for the facilities and the permission to publish. We thank the academic editor for the guidance during the review process. We are grateful to the two anonymous reviewers for meticulously going through the manuscript and for their useful suggestions.

## Author Contributions

**Conceptualization:** Ruchika B. Mohanty, Ratan Kar.

**Data curation:** Ruchika B. Mohanty, Amit K. Mishra, Kriti Mishra, Ratan Kar.

**Formal analysis:** Ruchika B. Mohanty, Amit K. Mishra, Kriti Mishra, Ratan Kar.

**Funding acquisition:** Ruchika B. Mohanty, Iswar C. Barua, Ratan Kar.

**Investigation:** Ruchika B. Mohanty, Amit K. Mishra, Kriti Mishra, Akhilesh K. Yadava, M. Firoze Quamar, Ratan Kar.

**Resources:** Ratan Kar.

**Software:** Amit K. Mishra, Akhilesh K. Yadava, M. Firoze Quamar.

**Supervision:** Iswar C. Barua, Ratan Kar.

**Validation:** M. Firoze Quamar, Iswar C. Barua, Ratan Kar.

**Writing – original draft:** Ruchika B. Mohanty, Ratan Kar.

**Writing – review & editing:** Ruchika B. Mohanty, Ratan Kar.

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
