## [Decision Letter · Decision Letter 0]

19 Jun 2023

PONE-D-22-33375Early onset of aridity in the past millennium: insights from vegetation dynamics and climate change in the alpine, cold-desert region of Trans Himalaya, IndiaPLOS ONE

Dear Dr. Kar,

Thank you for submitting your manuscript to PLOS ONE. After careful consideration, we feel that it has merit but does not fully meet PLOS ONE’s publication criteria as it currently stands. Therefore, we invite you to submit a revised version of the manuscript that addresses the points raised during the review process.

We look forward to receiving your revised manuscript.

Kind regards,

Huasheng Huang

Academic Editor

PLOS ONE

Journal Requirements:

"RBM is grateful to the Department of Science & Technology (DST), New Delhi for financial assistance under the Women Scientist Scheme (SR/WOS-A/EA-15/2019) and RK thanks DST for sponsoring a project under the Climate Change Programme (DST/CCP/PR/07/2011/G) under which field work and sampling were done."

"RBM is grateful to the Department of Science & Technology (DST), New Delhi for financial assistance under the Women Scientist Scheme (SR/WOS-A/EA-15/2019) and RK thanks DST for sponsoring a project under the Climate Change Programme (DST/CCP/PR/07/2011/G) "

"RBM is grateful to the Department of Science & Technology (DST), New Delhi for financial assistance under the Women Scientist Scheme (SR/WOS-A/EA-15/2019) and RK thanks DST for sponsoring a project under the Climate Change Programme (DST/CCP/PR/07/2011/G) "  

5. We note that Figure 1 in your submission contain map images which may be copyrighted. All PLOS content is published under the Creative Commons Attribution License (CC BY 4.0), which means that the manuscript, images, and Supporting Information files will be freely available online, and any third party is permitted to access, download, copy, distribute, and use these materials in any way, even commercially, with proper attribution. For these reasons, we cannot publish previously copyrighted maps or satellite images created using proprietary data, such as Google software (Google Maps, Street View, and Earth). For more information, see our copyright guidelines: http://journals.plos.org/plosone/s/licenses-and-copyright.

Reviewers' comments:

Reviewer's Responses to Questions

**Comments to the Author**

1. Is the manuscript technically sound, and do the data support the conclusions?

Reviewer #1: Yes

Reviewer #2: Yes

2. Has the statistical analysis been performed appropriately and rigorously? 

Reviewer #1: Yes

Reviewer #2: Yes

3. Have the authors made all data underlying the findings in their manuscript fully available?

Reviewer #1: Yes

Reviewer #2: Yes

4. Is the manuscript presented in an intelligible fashion and written in standard English?

Reviewer #1: Yes

Reviewer #2: Yes

5. Review Comments to the Author

Reviewer #1: Comments on the article entitled ‘Early onset of aridity in the past millennium: insights from vegetation dynamics and climate change in the alpine, cold-desert region of Trans Himalaya, India’. A good amount of palynology data has been generated from a challenging and inaccessible area of the Himalayas. The paper is quite well written but some modifications are suggested especially in the Introduction, Discussion and Conclusion sections. Some suggestions are as below:

1. The title has an emphasis on early onset of aridity in the Trans Himalayan region however, this is not very explicit in the text- results as well as discussion. The authors need to highlight the points based on which they have arrived to this finding. The discussion on early inception of aridity is very subtle and not very convincing. Changes need to be made in the text/ title accordingly.

2. Do the authors prefer to suggest early initiation of LIA and then put the terminal phase of MCA? The authors need to provide strong evidence as to why cannot it be interpreted as a period of aridity within MCA? The MCA is known for fluctuating precipitation and temperature conditions. Event based deposition or constant sedimentation rate?

3. The change in the climatic conditions from warm-moist to cold-arid has been made largely based on the percentage of broad-leaved taxa rather than the AP/NAP ratios. It may be helpful to include that detail in summary figure rather than AP/NAP ratio as these remain almost the same throughout the sediment profile.

4. Chronology is discussed in BP as well as AD and also centuries which makes it difficult and confusing for the reader to follow… uniformity is easy for correlation.

Some specific points-

Lines 15-16- this is not seen in the results. For more than half of the trench depth, the values of AP/NAP remain almost same.

Line 17- you mean lower values not lesser

Lines 22-24- “This cold-arid phase was, nevertheless, punctuated by a warm-moist period during 790 to 680 yr BP (AD 1160−1270), which marks the termination of the MCA. The full extent of the Little Ice Age (LIA) is well marked in the area.” Does this mean that the authors suggest beginning of LIA prior to termination of MCA?

Line 28- evidence

Line 29- late Holocene or last 1500 years?

Lines 30-31 delete.

Lines 56-73- proxies are known… not needed in introduction… can go to methods section if absolutely necessary.

Lines 86-88- do geographical boundaries matter?

Lines 109-110- refer Fig. 2a

Line 173- delete ‘.’ Before d.

Lines 220-222- not reflected in the results and figure.

Lines 280-292- Why is Pinus pollen the most abundant in surface as well as subsurface samples? Has this anything to do with the overall climatic conditions of the region?

Lines 294-300- The grain size is already mentioned in the description of three palynozones. Why repeat? Lot of it is methodology. This part along with the ternary diagram can be added to the methodology section.

Lines 327-329--- this is similar to present day-

Lines 452-498--- what do the authors want to highlight here? Are they discussing the time duration of the late Holocene climatic events?... there has never been any consensus as there are local factors governing. Such a discussion can be avoided.

Line 505- conclusion 3- reduced values? As compared to what?

Line 514- Add ‘( ‘ between BP and AD.

Line 518- Delete additional full stop.

Figure 10- shows overlap of chronology (present study) between DACP – MCA and MCA-LIA. Please check.

Reviewer #2: The present study investigates the samples collected from Hamtah Glacier, Lahaul-Spiti, India and provides deeper insights into the paleovegetation and paleoclimate of Trans Himalaya region. The 90 cm deep sedimentary sequence was chronologically assessed and found to expose the horizons dating back to 1580 yr BP. Based on the percentages of arboreal and non-arboreal pollen recovered from the different horizons, the authors divided the paleoclimatic profile into three phases. The three phases correspond to the globally recognized four major climatic events (DACP, MCA, LIA and CWP) of the Late Holocene. The episodes of cold-arid and warm-moist climate in the study area determined using a multi-proxy approach were further compared with the published records from other parts of the Himalaya and a few global sites. The study infers that the Late Holocene climatic events are asynchronous. The study further makes a huge contribution to science by providing evidence for the backwards extended period of LIA in the Trans Himalaya region.

The manuscript has been written really well. However, following are a few edits that I would suggest to make in the manuscript for a better reading:

Line 42: multi-proxy.

Line 49: Suggested paraphrase: The fluctuating intensities of the climate systems (ISM and WDs) have had variedly impacted the Himalayan and South Asian regions.

Line 51: Suggested paraphrase: the fusion of Higher or Greater Himalaya with Pir Panjal Range in north-western Himalaya forms a.....

Line 79: Use either sub-continent or subcontinent throughout the text.

Line 97: The initials of the abbreviation (SRTM) should be in upper case.

Line 120: Suggested paraphrase: Sporadic, low intensity ISM showers occur from July to September, whereas most of the precipitation falls as snowfall from November to March due to the WDs.

Line 135: Use either win-ward or windward throughout the text.

Line 188: Use either gm or g throughout the text, as per the journal's format.

Line 218: Suggested paraphrase: From bottom to top, three pollen zones (Pollen zone-I, Pollen zone-II and Pollen zone-III) have been demarcated in the sequence based on the increase or decrease in the percentages of arboreal and non-arboreal pollen (here represented as ratio of arboreal/non-arboreal pollen).

Line 246: Replace comma with the full stop and make the following text a separate sentence (However, Juniperus is present only in one sample.)

Line 263: Replace “usual” with “in the previous zone”

Line 286: ….in all the “surface as well as sub-surface” samples. Strike out the remaining text.

Line 286: Suggested paraphrase: Moreover, the high proportion of Pinus pollen, along with significant similarity in the palynological assemblages from both the realms enabled......

Line 321: The observed variations in the values of weathering indices throughout the trench reflect the changes in weathering intensity.

Line 336: ….the two can “be” observed.

Line 372: AD is written before the values. Maintain the format throughout the text.

Line 375: Suggested paraphrase: The maximum expansion of thermophilous broad-leaved forests from the Chandra Valley, around Kunzum La, during 1609−1303 yr BP has been correlated with the MCA.

Line 405: Close the parenthesis after Present.

Line 475: It should be 1300.

I have also attached the Pdf marked with suggestions for the authors' kind perusal.

6. PLOS authors have the option to publish the peer review history of their article (what does this mean?). If published, this will include your full peer review and any attached files.

Reviewer #1: No

Reviewer #2: No

---

## [Author Response · Author response to Decision Letter 0]

21 Aug 2023

Response to Reviewers

Academic editor

Response: We understand that the manuscript is according to PLOS ONE's style requirements.

Response: No permits were required regarding field access. The work was one of the components of the in-house projects of the Institute, which are already approved. We have acknowledged the Director (competent authority) of the Institute for the facilities provided and the permission to publish.

"RBM is grateful to the ………….(DST/CCP/PR/07/2011/G) under which field work and sampling were done."

Response: Have modified the Acknowledgements Section, as suggested. 

"RBM is grateful to the ……………….under the Climate Change Programme (DST/CCP/PR/07/2011/G) "

Response: We have removed the funding-related text from the manuscript and included our statement in the cover letter.

"RBM is grateful to the ………………under the Climate Change Programme (DST/CCP/PR/07/2011/G) " 

Response: We have stated the above in the cover letter.

5. We note that Figure 1 in your submission contain map images which may be copyrighted………

Response: May we emphatically state that Figure 1 is NOT copyrighted. Figure 1a & b were made using Corel Draw, while Figure 1c was made using ArcGIS 10.3 (also stated in the figure caption).

6. Please review your reference list…………

Response: No changes made in the reference list and also those cited in the manuscript.

Reviewer #1

We are grateful to the esteemed reviewer for meticulously going through the manuscript and for the very useful suggestions. We are pleased to see that he/she has found the paper ‘quite well written’ and for the encouraging comments. We have incorporated most of the suggestions; at few points we have differed and have given plausible explanations for the same. 

The point wise response to each query is given below: 

1. The title has an emphasis on early onset of aridity in the Trans Himalayan region however, this is not very explicit in the text- results as well as discussion. The authors need to highlight the points based on which they have arrived to this finding. The discussion on early inception of aridity is very subtle and not very convincing. Changes need to be made in the text/ title accordingly.

Response: Well, we would like to slightly differ here. An early onset of aridity in the Trans Himalayan region, at the beginning of the last millennium, is the novel finding of the present work. At 950 yr BP, there is a prominent decline in the AP/NAP ratio (from 82% to 65%), which clearly demarcates the two palynozones. This sharp fall in the frequency of arboreal pollen is the manifestation of the onset of cold-arid conditions, which are rather a little too early in this Trans Himalayan region, as compared to the other well-known records. We feel that we have provided robust data to back our inferences. Please see sections under Results and interpretations; sub-head Palynological analysis of the sub-surface sediments; Pollen zone-III; and also Discussion: 950 yr BP to the Present, 1st paragraph). Hope that we have provided satisfactory replies to the above queries and the esteemed reviewer would approve the same. 

Please also see our response to your point’s no. 2 & 3 below. 

2. Do the authors prefer to suggest early initiation of LIA and then put the terminal phase of MCA? The authors need to provide strong evidence as to why cannot it be interpreted as a period of aridity within MCA? The MCA is known for fluctuating precipitation and temperature conditions. Event based deposition or constant sedimentation rate?

Response: Yes, we are suggesting an early inception of LIA in the Trans-Himalayan region. You are right that ‘The MCA is known for fluctuating precipitation and temperature conditions.’ Nevertheless, this phase cannot be interpreted as a ‘period of aridity within MCA’, but rather as a different climatic phase, because the AP/NAP ratio registers the sharpest fall here (from 82% to 65%). Other than the AP/NAP ratio, the climatic distinction between this colder phase and the earlier warmer one is further supported by the proliferation of arid-loving Artemisia and lesser frequency of moist-loving ferns. Besides, the magnetic susceptibility and geochemical data also indicate a turnover from warmer to colder conditions. We have clearly presented the data for the same (please see Discussion: 950 yr BP to the Present, 1st paragraph). 

3. The change in the climatic conditions from warm-moist to cold-arid has been made largely based on the percentage of broad-leaved taxa rather than the AP/NAP ratios. It may be helpful to include that detail in summary figure rather than AP/NAP ratio as these remain almost the same throughout the sediment profile.

Response: We beg to differ somewhat! ‘The change in the climatic conditions from warm-moist to cold-arid’ has been done both on the basis of AP/NAP ratios, aided by the percentage of broad-leaved taxa. Your observation that the ‘AP/NAP ratio remain almost the same throughout the sediment profile,’ is for the lower half of the profile only (in the earlier two palynozones), where the AP/NAP ratio is almost similar (AP: 81% and 82%, respectively). However, these two zones can be distinguished by the change in the frequencies of broad-leaved elements (6% and 9%, respectively). Even subtle changes in the frequencies of the temperate broad-leaved taxa, which are thermophilic in nature, are considered good indicators of climatic changes in the Higher Himalayan region. Moreover, there is a sharp fall in the AP/NAP ratio between the second and third palynozones (AP: 82% and 65%, respectively). We have not put the changing frequencies of broad-leaved taxa in the summary diagram, but have rather discussed the above data comprehensively in the section under ‘Results and interpretations’; sub-head ‘Palynological analysis of the sub-surface sediments.’ 

4. Chronology is discussed in BP as well as AD and also centuries which makes it difficult and confusing for the reader to follow… uniformity is easy for correlation.

Response: We have given our ages in BP in the Chronology head. Most of the chronology has been discussed in BP, as cited in the respective works. Only in certain cases we have additionally given ages in AD in brackets, as these have appeared in the original publication, for correlation and clarity. Nonetheless, there were few minor discrepancies, which we have addressed.

Some specific points-

Lines 15-16- this is not seen in the results. For more than half of the trench depth, the values of AP/NAP remain almost same.

Response: We have answered the same in point no. 3 above. We had missed to mention about the importance of broad-leaved taxa here, so we added a few words regarding the same – ‘complemented by the frequencies of the broad-leaved taxa.’

Line 17- you mean lower values not lesser

Response: Yes.

Lines 22-24- “This cold-arid phase was, nevertheless, punctuated by a warm-moist period during 790 to 680 yr BP (AD 1160−1270), which marks the termination of the MCA. The full extent of the Little Ice Age (LIA) is well marked in the area.” Does this mean that the authors suggest beginning of LIA prior to termination of MCA?

Response: Yes, as pointed out earlier, the main premise of the paper is the early initiation of a cold-arid phase in the region, which is manifested by the longer duration of the LIA and well recorded by the different proxies used in the present study. Whereas the warmer phase, represented by the MCA, is not that prominent. Nonetheless, the palynological data has recorded a short warming episode between 790 and 680 yr BP, which we postulate as the culmination phase of the MCA, as it matches well with the ‘termination of the MCA’ as correlated with the global and Indian records.

Line 28- evidence

Response: Thanks for correcting.

Line 29- late Holocene or last 1500 years?

Response: Ok, the last 1580 years!

Lines 30-31 delete.

Response: Ok, fine!

Lines 56-73- proxies are known… not needed in introduction… can go to methods section if absolutely necessary.

Response: Beg your pardon, but we feel that these lines are better suited in the Introduction, where we introduce the tools (proxies) used in the present study. It is true that the different ‘proxies are known’; however, just a few lines about these may please be allowed. 

Lines 86-88- do geographical boundaries matter?

Response: Well, ‘geographical boundaries do not really matter’, but often the reviewers/editors ask for specifics about the area of study, specially for the benefit of non Indian readers. So we think that a couple of lines are OK.

Lines 109-110- refer Fig. 2a

Response: Done, as suggested.

Line 173- delete ‘.’ Before d.

Response: Yes.

Lines 220-222- not reflected in the results and figure.

Response: I think we have addressed similar queries earlier – please refer to point no. 3 and your query regarding Lines 15-16. Besides, Reviewer #2 has suggested rephrasing the same sentences, which we have complied with.

Lines 280-292- Why is Pinus pollen the most abundant in surface as well as subsurface samples? Has this anything to do with the overall climatic conditions of the region?

Response: Thanks for pointing this out; we had missed to write reasons for the abundance of Pinus pollen. We have extensively worked in the Himalayan region and everywhere we have encountered the predominance of Pinus pollen in the palynological assemblages (References already given: 26, 27). We have added the following lines in the text:

Pinus spp. are prolific pollen producers and owing to their excellent buoyancy, are efficiently transported by winds to long distances. They also have good preservation potential in the sediments, which enables them to be bountifully recorded in the palynological assemblages. It has been observed that even if Pinus is not growing in the immediate vicinity of the sampling locations, its pollen can be predominant to such an extent that it completely overwhelms the other pollen taxa in the palynological records.

Lines 294-300- The grain size is already mentioned in the description of three palynozones. 

Why repeat? Lot of it is methodology. This part along with the ternary diagram can be added to the methodology section.

Response: It is not really a repetition. In the Methodology section, the chemical processing of the samples and instrument involved were briefly mentioned. Whereas here, a few lines about the utility and need for grain size analysis and the results obtained thereby, have been given.

Lines 327-329--- this is similar to present day-

Response: Perhaps you mean that the AP/NAP ratio is similar to the present day (CWP)? If that is so, then it is just a coincidence. We have faithfully presented our data.

Lines 452-498--- what do the authors want to highlight here? Are they discussing the time duration of the late Holocene climatic events?... there has never been any consensus as there are local factors governing. Such a discussion can be avoided.

Response: Yes, your observations are correct that as far as the late Holocene climatic events are concerned “there has never been any consensus as there are local factors governing.” However, do allow us to say here that there was lack of global contextualisation of the Indian records, as the earlier works were not of good resolution. Nonetheless, during the last two decades, or so, good high-resolution multi-proxy, Holocene climatic records have been generated from India. Therefore, our attempt is to highlight the Indian data and correlate these with the well-known global records, so as to have global connotation of the Indian works. Hope you would agree to our point of view.

Line 505- conclusion 3- reduced values? As compared to what?

Response: Yes, we agree that it is confusing, so we have deleted the said part and corrected the sentence.

Line 514- Add ‘( ‘ between BP and AD.

Response: Right, thank you.

Line 518- Delete additional full stop.

Response: Yes.

Figure 10- shows overlap of chronology (present study) between DACP – MCA and MCA-LIA. Please check.

Response: Yes, the ‘overlap’ is due to the asynchronicity of the Late Holocene climatic events: DACP. MCA, LIA and CWP regarding their inception and duration, which the esteemed reviewer was referring to earlier. However, we wanted to show the Late Holocene climatic events from India in a single figure, along with the present work and the important Global records. 

Reviewer #2

We are indeed thankful to the learned reviewer for sparing his/her valuable time and for rectifying our errors. We are pleased to note that he/she has appreciated our efforts, and supported our inferences, especially the ‘backward extended period of LIA in the Trans Himalaya region.’

We have addressed all the suggestions/corrections, which are as under:

Line 42: multi-proxy.

Response: Rectified

Line 49: Suggested paraphrase: The fluctuating intensities of the climate systems (ISM and WDs) have had variedly impacted the Himalayan and South Asian regions.

Response: Done as suggested, thank you!

Line 51: Suggested paraphrase: the fusion of Higher or Greater Himalaya with Pir Panjal Range in north-western Himalaya forms a.....

Response: Rectified accordingly.

Line 79: Use either sub-continent or subcontinent throughout the text.

Response: Ok

Line 97: The initials of the abbreviation (SRTM) should be in upper case.

Response: Yes, thank you!

Line 120: Suggested paraphrase: Sporadic, low intensity ISM showers occur from July to September, whereas most of the precipitation falls as snowfall from November to March due to the WDs.

Response: Ok, fine

Line 135: Use either win-ward or windward throughout the text.

Response: Yes

Line 188: Use either gm or g throughout the text, as per the journal's format.

Response: OK

Line 218: Suggested paraphrase: From bottom to top, three pollen zones (Pollen zone-I, Pollen zone-II and Pollen zone-III) have been demarcated in the sequence based on the increase or decrease in the percentages of arboreal and non-arboreal pollen (here represented as ratio of arboreal/non-arboreal pollen).

Response: Rephrased as suggested, thanks for the same.

Line 246: Replace comma with the full stop and make the following text a separate sentence (However, Juniperus is present only in one sample.)

Response: Done accordingly

Line 263: Replace “usual” with “in the previous zone”

Response: Ok

Line 286: ….in all the “surface as well as sub-surface” samples. Strike out the remaining text.

Response: Rectified as suggested

Line 286: Suggested paraphrase: Moreover, the high proportion of Pinus pollen, along with significant similarity in the palynological assemblages from both the realms enabled......

Response: Rephrased as suggested

Line 321: The observed variations in the values of weathering indices throughout the trench reflect the changes in weathering intensity.

Response: Rectified accordingly

Line 336: ….the two can “be” observed.

Response: Yes

Line 372: AD is written before the values. Maintain the format throughout the text.

Response: Yes

Line 375: Suggested paraphrase: The maximum expansion of thermophilous broad-leaved forests from the Chandra Valley, around Kunzum La, during 1609−1303 yr BP has been correlated with the MCA.

Response: Rephrased as suggested

Line 405: Close the parenthesis after Present.

Response: Yes

Line 475: It should be 1300.

Response: Yes, thank you

I have also attached the Pdf marked with suggestions for the authors' kind perusal.

Response: Thanks for the same; we have addressed all the markings.

---

## [Editor Report · Decision Letter 1]

24 Aug 2023

PONE-D-22-33375R1Early onset of aridity in the past millennium: insights from vegetation dynamics and climate change in the alpine, cold-desert region of Trans Himalaya, IndiaPLOS ONE

Dear Dr. Kar,

Thank you for submitting your manuscript to PLOS ONE. After careful consideration, we feel that it has merit but does not fully meet PLOS ONE’s publication criteria as it currently stands. Therefore, we invite you to submit a revised version of the manuscript that addresses the points raised during the review process.

We look forward to receiving your revised manuscript.

Kind regards,

Huasheng Huang

Academic Editor

PLOS ONE

Journal Requirements:

Additional Editor Comments:

In "Revised Manuscript with Track Changes" (lines 137-143 and 267), I only saw two track changes? Please indicate all your track changes and submit again.

---

## [Author Response · Author response to Decision Letter 1]

28 Aug 2023

‘This is in response to your recent mail dated 25.08.2023 regarding your Additional Editor Comments: In "Revised Manuscript with Track Changes" (lines 137-143 and 267), I only saw two track changes? Please indicate all your track changes and submit again’. May I humbly submit -

That I had addressed all your and that of the two reviewers queries and had sent those in two separate files labeled 'Revised Manuscript with Track Changes' and 'Manuscript', along with the ‘rebuttal letter’. 

After a few days, I received another mail from Edrian Nim Tolentino, asking me to provide additional information regarding the permits obtained for the work to be added in the Methods section. 

I was under the impression that my earlier submission, in response to your Editorial observations and the reviewers queries, have been taken care of, hence I modified only that part in the Methods section that Edrian had desired and sent it back. Therefore, you could see only few track changes and hence the confusion. Anyway, I am now submitting the revised manuscript addressing both yours and Edrian’s observations. Hope you will bear with us.

---

## [Decision Letter · Decision Letter 2]

29 Sep 2023

PONE-D-22-33375R2Early onset of aridity in the past millennium: insights from vegetation dynamics and climate change in the alpine, cold-desert region of Trans Himalaya, IndiaPLOS ONE

Dear Dr. Kar,

Thank you for submitting your manuscript to PLOS ONE. After careful consideration, we feel that it has merit but does not fully meet PLOS ONE’s publication criteria as it currently stands. Therefore, we invite you to submit a revised version of the manuscript that addresses the points raised during the review process.

We look forward to receiving your revised manuscript.

Kind regards,

Huasheng Huang

Academic Editor

PLOS ONE

Journal Requirements:

Additional Editor Comments:

There is still a reviewer that is not satisfied with your revision. Please revise based on his comments.

Reviewers' comments:

Reviewer's Responses to Questions

**Comments to the Author**

1. If the authors have adequately addressed your comments raised in a previous round of review and you feel that this manuscript is now acceptable for publication, you may indicate that here to bypass the “Comments to the Author” section, enter your conflict of interest statement in the “Confidential to Editor” section, and submit your "Accept" recommendation.

Reviewer #1: (No Response)

Reviewer #2: All comments have been addressed

2. Is the manuscript technically sound, and do the data support the conclusions?

Reviewer #1: Yes

Reviewer #2: Yes

3. Has the statistical analysis been performed appropriately and rigorously? 

Reviewer #1: Yes

Reviewer #2: Yes

4. Have the authors made all data underlying the findings in their manuscript fully available?

Reviewer #1: Yes

Reviewer #2: Yes

5. Is the manuscript presented in an intelligible fashion and written in standard English?

Reviewer #1: Yes

Reviewer #2: Yes

6. Review Comments to the Author

Reviewer #1: I am not convinced with the responses to my comments on the article and the responses also appear to be rather combative than professional. The authors have probably not understood the the issues I had pointed out and therefore have not been able to adequately address the same. I would suggest the authors to revisit my earlier comments related to the ' early onset of aridity', 'the duration of LIA' and "the termination of MCA" and resolve the discrepancies.

Reviewer #2: The authors have revised the manuscript very efficiently and effectively. I have no other questions or concerns left. The manuscript is ready to be published now.

7. PLOS authors have the option to publish the peer review history of their article (what does this mean?). If published, this will include your full peer review and any attached files.

Reviewer #1: No

Reviewer #2: No

---

## [Author Response · Author response to Decision Letter 2]

24 Nov 2023

Response to Reviewers

Reviewer # 1

At the outset, may we most humbly submit that if our responses ‘have appeared to be rather combative than professional’, then we do beg your pardon. Perhaps, in our attempt to profess our points of view, we might have been somewhat overzealous. Hope you will bear with us and excuse us for the same. 

As suggested, we revisited and had a look at the earlier comments and we sincerely believe that we had addressed the queries to the best of our abilities. Nonetheless, we would surely like to further resolve the issues and address your main concerns related to:

Early onset of aridity: Please allow us to say that in our opinion we have the required palynological data, backed by mineral magnetic and geochemical proxies to propose an early onset of cold amd arid conditions in the region. At 950 yr BP we find:

a. A sharp decline in the AP/NAP ratio (from 82% to 65%).

b. Proliferation of arid-loving steppe elements, specially Artemesia, and a consequent decline in moist-loving ferns.

c. Low magnetic susceptibility, indicating less weathering as a result of prevailing aridity.

d. Increase in geochemical indices, implying cold-arid conditions.

We have discussed these findings comprehensively under Results and interpretations; sub-head Palynological analysis of the sub-surface sediments; Pollen zone-III; and also Discussion: 950 yr BP to the Present, 1st paragraph. 

Also, as suggested by you in your earlier comment that ‘as to why cannot it be interpreted as a period of aridity within MCA? Yes, your observation is fine and we have incorporated it in the text as well (please see Discussion, 1330 to 950 yr BP, first paragraph). But even then, aridity has crept in early in the region, during the MCA itself.

Besides, your earlier observation that – “Does this mean that the authors suggest beginning of LIA prior to termination of MCA? So taking your suggestion forward, we understand that the LIA had begun only after the termination of the MCA. Required modifications have been done in the text (Abstract and Discussion). Also we are further discussing the issue below:

Duration of LIA and the termination of MCA: We understand that there might have been some confusion regarding the duration of LIA vis-à-vis the termination of MCA, in the way we had presented these in our paper. We have now clearly stated that the LIA commences from 680 yr BP (AD 1270), after the termination of the MCA (please note that the commencement of the LIA at 680 yr BP (AD 1270) in the present region, is rather early if compared with the generally established global records (LIA: AD 1500 to 1850). Suitable modifications have been made in the text (please see Discussion, 950 yr BP to the Present, second paragraph; Conclusion). 

We earnestly hope that we have now addressed the main issues and have been able to resolve the discrepancies. Looking forward to your benevolent support on our findings.

---

## [Editor Report · Decision Letter 3]

30 Nov 2023

Early onset of aridity in the past millennium: insights from vegetation dynamics and climate change in the alpine, cold-desert region of Trans Himalaya, India

PONE-D-22-33375R3

Dear Dr. Kar,

We’re pleased to inform you that your manuscript has been judged scientifically suitable for publication and will be formally accepted for publication once it meets all outstanding technical requirements.

Kind regards,

Huasheng Huang

Academic Editor

PLOS ONE
---

## [Editor Report · Acceptance letter]

13 Dec 2023

PONE-D-22-33375R3 

PLOS ONE

Dear Dr. Kar, 

I'm pleased to inform you that your manuscript has been deemed suitable for publication in PLOS ONE. Congratulations! Your manuscript is now being handed over to our production team.

Kind regards, 

on behalf of

Dr. Huasheng Huang 

Academic Editor

PLOS ONE